# A neuroligin-2-YAP axis regulates progression of pancreatic intraepithelial neoplasia

Emanuele Middonti [ID][1,2✉], Elena Astanina[1,2], Edoardo Vallariello [ID][1,2], Roxana Maria Hoza[1,2], Jasna Metovic[1], Rosella Spadi [ID][3], Carmen Cristiano[3], Mauro Papotti[1,4], Paola Allavena [ID][5], Francesco Novelli[6,7,8], Sushant Parab[1,2], Paola Cappello [ID][6,7,8], Aldo Scarpa[9,10], Rita Lawlor[9,10], Massimo Di Maio[1,11], Marco Arese[1,2,12] & Federico Bussolino [ID][1,2,12✉]

## Abstract

**Pancreatic ductal adenocarcinoma (PDAC) is a tumor with a dismal prognosis that arises from precursor lesions called pancreatic intraepithelial neoplasias (PanINs). Progression from low- to high-grade PanINs is considered as tumor initiation, and a deeper understanding of this switch is needed. Here, we show that synaptic molecule neuroligin-2 (NLGN2) is expressed by pancreatic exocrine cells and plays a crucial role in the regulation of contact inhibition and epithelial polarity, which characterize the switch from low- to high-grade PanIN. NLGN2 localizes to tight junctions in acinar cells, is diffusely distributed in the cytosol in low-grade PanINs and is lost in high-grade PanINs and in a high percentage of advanced PDACs. Mechanistically, NLGN2 is necessary for the formation of the PALS1/PATJ complex, which in turn induces contact inhibition by reducing YAP function. Our results provide novel insights into NLGN2 functions outside the nervous system and can be used to model PanIN progression.**

**Keywords** Neuroligin; YAP; Pancreatic Intraepithelial Neoplasia; Cell Polarity; Contact Inhibition
**Subject Categories** Cancer; Cell Adhesion, Polarity & Cytoskeleton; Signal Transduction

## Introduction

Pancreatic ductal adenocarcinoma (PDAC) is the most prevalent pancreatic neoplasm and is considered a fatal incurable disease mainly due to its late diagnosis (Ryan et al, 2014). Therefore, more efforts to shed light on the initial stages of the disease are needed.

Oncogenic mutations in *K-RAS* are the driving events that lead to the transformation from normal pancreatic tissues to PDAC. In pancreatic acinar cells, *K-RAS* overactivation induces transdifferentiation into duct-like intermediates (acinar-to-ductal metaplasia, ADM) (Guerra et al, 2007), which evolve into precursor lesions named pancreatic intraepithelial neoplasias (PanINs). PanINs are classified as low- and high-grade according to the degree of dysplasia (Basturk et al, 2015) and progress to PDAC by accumulating alterations in driver genes, including *CDKN2A*, *TP53*, and *SMAD4* (Ryan et al, 2014). Low-grade PanINs occur in a fraction of patients with nonmalignant pancreatic diseases, and the percentage of cases progressing to high-grade PanIN and PDAC is ~1.4% (Peters et al, 2018). This transition requires decades and is fostered by pancreatitis (Makohon-Moore and Iacobuzio-Donahue, 2016; Murphy et al, 2013).

High-grade PanINs represent authentic PDAC onset (Makohon-Moore and Iacobuzio-Donahue, 2016; Murphy et al, 2013) and are characterized by nuclear abnormalities, increased cell proliferation, and loss of cell polarity. Although these features are recognized as general characteristics of epithelial cancers (Halaoui and McCaffrey, 2015; Pease and Tirnauer, 2011), studies have generally focused on established PDAC, and information on proliferation and polarity mechanics in precursor lesions of the disease is negligible.

Another aspect characterizing PanIN evolution is neuronal plasticity, an early event of PDAC onset contributing to its progression by perineural invasion (Biankin et al, 2012; Schorn et al, 2017; Sinha et al, 2017; Stopczynski et al, 2014). Interestingly, molecules involved in neuronal plasticity are often deregulated in PDAC and in genetically engineered mouse models (GEMMs) in which *K-Ras* and *Trp53* mutations are the sole driving events (Stopczynski et al, 2014).

Neuroligins (NLGNs) are postsynaptic adhesion proteins involved in synapse maturation. The proteins encoded by five and three NLGN genes in humans and mice, respectively, bind to

[1]Department of Oncology, University of Torino, 10043 Orbassano, Italy. [2]Candiolo Cancer Institute-IRCCS-FPO, 10060 Candiolo, Italy. [3]SC Oncologia Medica, Città della Salute e della Scienza di Torino, 10126 Torino, Italy. [4]Division of Pathology at Città della Salute e della Scienza di Torino, 10126 Torino, Italy. [5]IRCCS, Humanitas Clinical and Research Center, 20089 Rozzano, Italy. [6]Department of Molecular Biotechnology and Health Sciences, University of Torino, 10126 Torino, Italy. [7]Laboratory of Tumor Immunology, Center for Experimental Research and Medical Studies, Città della Salute e della Scienza di Torino, 10126 Torino, Italy. [8]Molecular Biotechnology Center, University of Torino, 10125 Torino, Italy. [9]Applied Research Center (ARC-NET), University of Verona, 37134 Verona, Italy. [10]Department of Diagnostics and Public Health, University of Verona, 37134 Verona, Italy. [11]Medical Oncology, Ordine Mauriziano Hospital, 10128 Torino, Italy. [12]These authors contributed equally: Marco Arese, Federico Bussolino. ✉E-mail: Emanuele.middonti@unito.it; Federico.Bussolino@unito.it

presynaptic neurexins and form an asymmetric bridge between the pre- and postsynaptic membranes, contributing to the polarized features of synapses (Craig and Kang, 2007). Via their intracellular domain, NLGNs recruit scaffold proteins and neurotransmitter receptors (Poulopoulos et al, 2009; Soykan et al, 2014).

Interestingly, as documented in Drosophila, NLGNs share evolutionary origins with tight junctions, which are instrumental in epithelial polarity (Schulte et al, 2003), and accumulating evidence indicates that NLGNs are also expressed outside of the nervous system (Bottos et al, 2009; Shah et al, 2023; Suckow et al, 2008; Zhang et al, 2013).

It has been reported that NLGN2 binds to PATJ (Kurschner et al, 1998), a component of the Crumbs (CRB) polarity complex also containing CRB, MALS-3, and PALS1, which localizes to the apical region of the cell in proximity to tight junctions. Polarity cues can regulate the Hippo pathway, a serine/threonine kinase cascade that converges on LATS and MST, which in turn phosphorylates YAP and TAZ, blocking their nuclear translocation. In the nucleus, YAP and TAZ interact with TEAD, initiating a genetic program whose effects include the regulation of cell proliferation, epithelial-mesenchymal transition, and cancer stem cells self-renewal, such as in breast cancer where the loss of the cell polarity determinant Scribble upregulates TAZ activity (Corde-nonsi et al, 2011). Recent findings indicate that the CRB complex senses and regulates cell polarity and leads to control of proliferation through the Hippo pathway too (Chen et al, 2010; Mao et al, 2017; Szymaniak et al, 2015; Varelas et al, 2010). Mechanistically, the CRB module recruits YAP through the AMOT adapter, reducing YAP availability in the cytosol and promoting its interaction with LATS kinase (Moleirinho et al, 2017).

The aim of the present study was to identify key mechanisms that drive the transition from low- to high-grade PanIN and then to PDAC. We find that NLGN2 is expressed in the normal pancreas and exhibits reduced expression in PanINs, with a marked decrease in high-grade PanINs, and in most PDAC cases. Consistent with NLGN2 expression in low-grade PanINs and in a small percentage of PDACs, we find that NLGN2 expression correlates with cell polarity and proliferation. Furthermore, we provide evidence that NLGN2 loss leads to aberrant epithelial cystogenesis and proliferation in confluent cells in vitro. We show that NLGN2 drives cell polarization through regulation of the PALS1/PATJ complex and that its silencing induces loss of contact inhibition via over-activation of YAP.

# Results

## Nlgn2 is expressed in the normal pancreas and is progressively downregulated in PanINs

We established a stable PanIN cell line from pancreatic cells of *K-Ras$^{+/LSLG12Vgeo}$;Elas-tTA/tetO-Cre* (*Elast-K-Ras$^{G12V}$*) mice (EK cells), which spontaneously develop ADM and PanINs from acinar cells and rarely develop PDAC (Guerra et al, 2007). Cells isolated from 12-month-old mice with high-grade PanINs (Fig. EV1A) were stable for up to 20 in vitro passages, suggesting that they overcame *K-Ras$^{G12V}$*-induced senescence, a typical feature of low-grade PanINs (Guerra et al, 2011). The cells showed a mesenchymal phenotype, and when cultured in 3D Matrigel, they formed

aberrant and unpolarized structures consistent with the presence of mutated *K-Ras* (Magudia et al, 2012). The acinar origin of EK cells in mice was confirmed by the presence of β-galactosidase activity as a surrogate marker for Cre recombinase activity and the expression of *LacZ*, which is coexpressed with *K-Ras$^{G12V}$* (Guerra et al, 2007) (Fig. EV1B).

Our attempts to obtain control acinar cells from *Elast-K-Ras$^{+/+}$* mice were unsuccessful because of the high number of contaminant cells, the presence of cells with a short lifespan, and the occurrence of spontaneous acinar-to-ductal metaplasia (Hall and Lemoine, 1992; Rooman et al, 2000), therefore, we chose the murine 266-6 immortalized acinar cell line (Ornitz et al, 1985). RNA seq data from 266-6 and EK cells corroborate the metaplastic nature of EK, as indicated by the expression of ductal Keratins and the downregulation of the acinar Amylases. Furthermore, these cells did not express Ins, Gcg, or Gfap, allowing us to exclude the possibility that endocrine and stellate cells were present (links in the Data availability section). Interestingly, GSEA analysis of RNA seq data revealed gene sets related to neuronal pathways enriched in 266-6 (Figs. 1A and EV1C), suggesting that such genetic programs sustain acinar cell functions and that their deregulation favors PanIN formation. Accordingly, with observations that synapses and epithelial cells share common features regarding the polarization of cell membranes and compartments (Burute and Kapitein, 2019; Simons et al, 1992), the gene set "protein–protein interactions at synapses", which includes trans-synaptic adhesion molecules *Nrxn2* and *Nlgn2*, was enriched in 266-6 (NES = −1.97, $p < 0.001$; Figs. 1A and EV1D). *Nlgn2* transcript exhibited the highest levels among *Neuroligins* in 266-6 cells (Fig. EV1E) and both mRNA and protein dramatically decreased in EK cells (Fig. 1A).

The in vitro expression of NLGN2 was further validated in *Elast-K-Ras$^{G12V}$* mice and human pancreatic tissues. NLGN2 was found to be widely expressed in different pancreas cell types, such as ductal cells and pancreatic islets, in both mouse and human samples, as already reported (Fig. EV2A). In *Elast-K-Ras$^{+/+}$*, *Elast-K-Ras$^{G12V}$*, and human normal acinar (Fig. 1B) and ductal cells (Fig. EV2B), NLGN2 was abundantly expressed, while its expression was slightly reduced and distributed diffusely in the cytosol of low-grade PanINs and was drastically downregulated in high-grade PanINs (Fig. 1B). Despite the widespread NLGN2 downregulation observed in these lesions, a small subset of cells with high NLGN2 expression was detected within low- and high-grade PanINs in both mouse and human samples (Figs. 1B and EV2C). In these cells, as well as in duodenal tuft cells (Gerbe et al, 2009), NLGN2 was coexpressed with doublecortin-like kinase 1 (DCLK1) (Fig. EV2C), an interactor of NLGN2 (Kang et al, 2014) and a stemness marker in several cancers, including pancreatic preinvasive lesions, and in the gastrointestinal tract (Bailey et al, 2014; Westphalen et al, 2014). Although these NLGN2$^{high}$ cells represent an interesting phenom-enon, they require separate specific studies, while here, we focused on the NLGN2 downregulation observed during PanIN evolution.

## Nlgn2 expression reflects changes in PanIN progression

In ductal and acinar cells, NLGN2 exhibited a polarized apical colocalization with the tight junction marker ZO1 (Figs. 1C and EV2B), with superimposable appearance between normal cells of *Elast-K-Ras$^{+/+}$* mice and untransformed cells of *Elast-K-Ras$^{G12V}$*

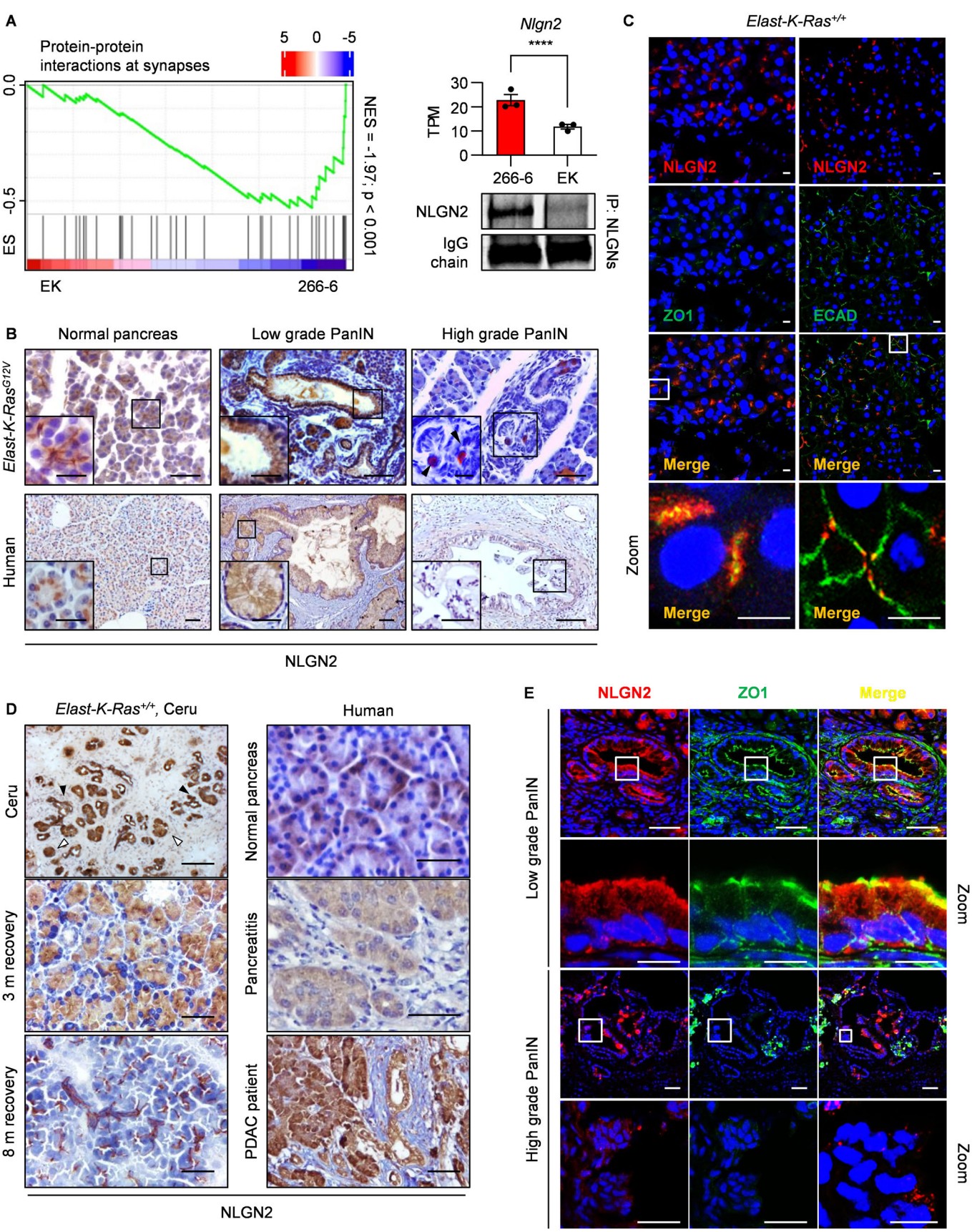

◀  **Figure 1.  Nlgn2 is expressed in the normal pancreas, and its expression is reduced in high-grade PanINs.**

(A) Left panel: negative enrichment of "protein–protein interactions at synapses" gene set in EK vs 266-6 cells. The red and blue color scale indicate $log_2$ mRNA fold change. ES: enrichment score; NES: normalized enrichment score. Right: *Nlgn2* expression is lower in EK cells than in 266-6 cells, as shown by TPM counts in RNA seq analysis (top) and Western blotting of NLGN2 in pan-NLGN immunoprecipitates (bottom). The primary antibody IgG light chain used to immunoprecipitate NLGNs is shown for normalization. (B) Immunohistochemical analysis of NLGN2 expression in *Elast-K-Ras$^{G12V}$* mice, normal human pancreatic tissues and PanINs. Images of mouse PanINs are representative of low and high grades at 4 and 12 months of age. The insets show higher-magnification images. The arrowheads in the insets indicate NLGN2-expressing cells in high-grade PanINs. (C) Immunofluorescence analysis of NLGN2 localization in the pancreas of 6-month-old mice. The images are representative of acinar cells of 6 *Elast-K-Ras$^{+/+}$* and are superimposible with untransformed acinar cells of 6 *Elast-K-Ras$^{G12V}$* mice. NLGN2 and ZO1 are colocalized in the apical portion of pancreatic acini (left), while NLGN2 and ECAD are distributed in distinct domains with only a small area of colocalization (right). The bottom panels are a higher magnification of the upper images (Zoom). (D) NLGN2 localization in acinar cells is altered by inflammation. Immunohistochemical analysis shows diffuse NLGN2 localization in acinar cells in pancreatitis foci in *Elast-K-Ras$^{+/+}$* mice. The left panel shows from top to bottom: cytosolic NLGN2 diffusion in *Elast-K-Ras$^{+/+}$* mouse ADM tissue (black arrowheads) and acini (white arrowheads) treated with caerulein for 3 months; cytosolic NLGN2 diffusion in *Elast-K-Ras$^{+/+}$* acini 3 months after caerulein cessation; restoration of NLGN2 apical distribution 8 months after caerulein cessation. Right panel, from top to bottom: NLGN2 localization in normal human pancreas; abundant and diffuse NLGN2 expression in human pancreatitis samples; acinar cells in PDAC patients showing diffuse NLGN2 staining. (E) Immunofluorescence analysis of NLGN2 and ZO1 expression in low- and high-grade PanINs in *Elast-K-Ras$^{G12V}$* mice. In low-grade PanINs, NLGN2 was occasionally colocalized with apical ZO1. In unpolarized high-grade PanINs, the expression of both NLGN2 and ZO1 was reduced, with no area of colocalization. Higher magnification are shown (Zoom). Data information: in (A) left: gene set is identified by GSEA (Kolmogorov–Smirnov test). Right: mean TPM counts ± SEM; *n* of biological replicates = 3; Wald test: adjusted *p* value = 3.03E-24: ****. (B) Mice scale bar: 0.05 mm, images are representative of 12 mice; human scale bar: 0.1 mm; Inset: *Elast-K-Ras$^{G12V}$* mice scale bar: 0.02 mm, human samples scale bar: 0.05 mm. (C) Scale bar: 20 µm. (D) Images from patients (scale bar: 0.2 mm) and caerulein-treated mice (scale bar: 0.05 mm) are representative of three samples. (E) Scale bar: 100 µm; Zoom low-grade PanINs and Zoom high-grade PanINs "Merge" scale bar: 25 µm; Zoom high-grade PanINs NLGN2 and ZO1 scale bar: 50 µm. Images are representative of 12 mice. Source data are available online for this figure.

mice. Although NLGN2 was apparently colocalized with the *adherens* junction marker ECAD, a more accurate imaging analysis showed only a partial overlap and exclusion from the basolateral domain (Fig. 1C).

In control *Elast-K-Ras$^{+/+}$* mice, caerulein-induced pancreatitis (Guerra et al, 2007) promoted intracellular localization of NLGN2 in acinar cells and ADM without reducing its expression level (Fig. 1D). Of note, 8 months after caerulein withdrawal, apical localization was restored (Fig. 1D). This observation paralleled the findings observed in human pancreatitis and in normal acinar cells adjacent to human adenocarcinoma foci (Fig. 1D), where apical expression of NLGN2 was lost. NLGN2 was also localized to the apical domain and in proximity to tight junctions in low-grade PanINs (Fig. 1E), although it exhibited a partial diffuse localization in the cytosol or localization to the basolateral region. In high-grade PanINs, NLGN2 expression was consistently reduced and dispersed in the cytosol (Fig. 1E).

## Prognostic significance of Nlgn2 in PDAC

Because *Elast-K-Ras$^{G12V}$* mice rarely develop tumors, we exploited the Metabolic Gene Rapid Visualizer database (MERAV; Shaul et al, 2016) to analyze NLGN2 expression in human PDAC. As shown in Fig. 2A, *NLGN2* expression was significantly lower in PDAC (57 cases) than in normal tissue (16 cases). The specificity of this analysis was supported by the downregulation of the acinar and insular cell markers *AMY2B* and *INS*, while the expression of the tumor marker *KRT19* was increased (Fig. EV3A). NLGN2 loss in PDAC was validated in *Pdx1-Cre*; *K-Ras$^{+/LSLG12D}$* (*Pdx1-K-Ras$^{G12D}$*) mice, which spontaneously develop PDAC between 4 and 6 months of age (Hingorani et al, 2003). As in *Elast-K-Ras$^{G12V}$* mice, *Pdx1-K-Ras$^{G12D}$* mice exhibited apical NLGN2 expression in pancreatic acini and in low-grade PanINs but no NLGN2 expression in high-grade PanINs, similar to the pattern observed in *Elast-K-Ras$^{G12V}$* mice. Interestingly, tumors exhibited heterogeneous NLGN2 expression, with most PDACs negative for NLGN2 expression and rare exceptions in differentiated, polarized glands that retained NLGN2 (Fig. 2B).

We confirmed this heterogeneous expression pattern in human PDAC by analyzing the expression of NLGN2 in 204 samples. We considered 135 samples to be NLGN2-negative due to the very weak or absent immunohistochemical signals, and these samples were from morphologically unpolarized tumors. In the 69 positive samples, NLGN2 exhibited different localization patterns in the patients, with minor differences within the same tumor. The localization patterns included a cell contact and apical distribution in differentiated and polarized glands, and an intracellular diffuse distribution in less polarized tumors (Fig. 2B).

To address whether the observed heterogeneous expression of NLGN2 in PDAC is clinically relevant, we examined the expression of the proliferation marker Ki67 (Zinczuk et al, 2018). We analyzed 14 and 11 human PDAC samples with negative and positive NLGN2 expression, respectively, and found a significant inverse correlation between the level of NLGN2 and the percentage of nuclear Ki67 (Fig. 2C). To better define the relevance of the above observation, we included the 204 patients in whose tumor samples NLGN2 expression was analyzed in survival analysis. As detailed in Table 1, there were no significant differences between the NLGN2-negative and NLGN2-positive patients in terms of major prognostic factors (clinical stage, infiltration of surgical margins) or in terms of administration of perioperative (neoadjuvant/adjuvant) chemotherapy.

The median survival time was longer in patients with positive NLGN2 expression (34.7 months, 95% confidence interval (CI) 17.1–52.3) than in patients with negative NLGN2 expression (25.2 months, 95% CI 20.6–29.7) (Fig. 2D). The survival probabilities for NLGN2-positive patients vs. NLGN2-negative patients were 85.5 vs. 83.0% at 1 year (*p* = 0.636), 62.1 vs. 52.3% at 2 years (*p* = 0.179), 49.3 vs. 34.3% at 3 years (*p* = 0.045), 43.7 vs. 27.7% at 4 years (*p* = 0.033), and 35.8 vs. 20.9% at 5 years (*p* = 0.055) (Fig. 2D).

A similar result was found in the multivariable analysis, in which NLGN2 expression, clinical stage, infiltration of surgical margins, and administration of perioperative chemotherapy were included. The presence of positive margins (hazard ratio 1.55, 95% CI 1.06–2.27, *p* = 0.024) was associated with worse survival, and administration of chemotherapy was associated with better survival (hazard ratio 0.51, 95% CI 0.33–0.78, *p* = 0.002). Negative NLGN2

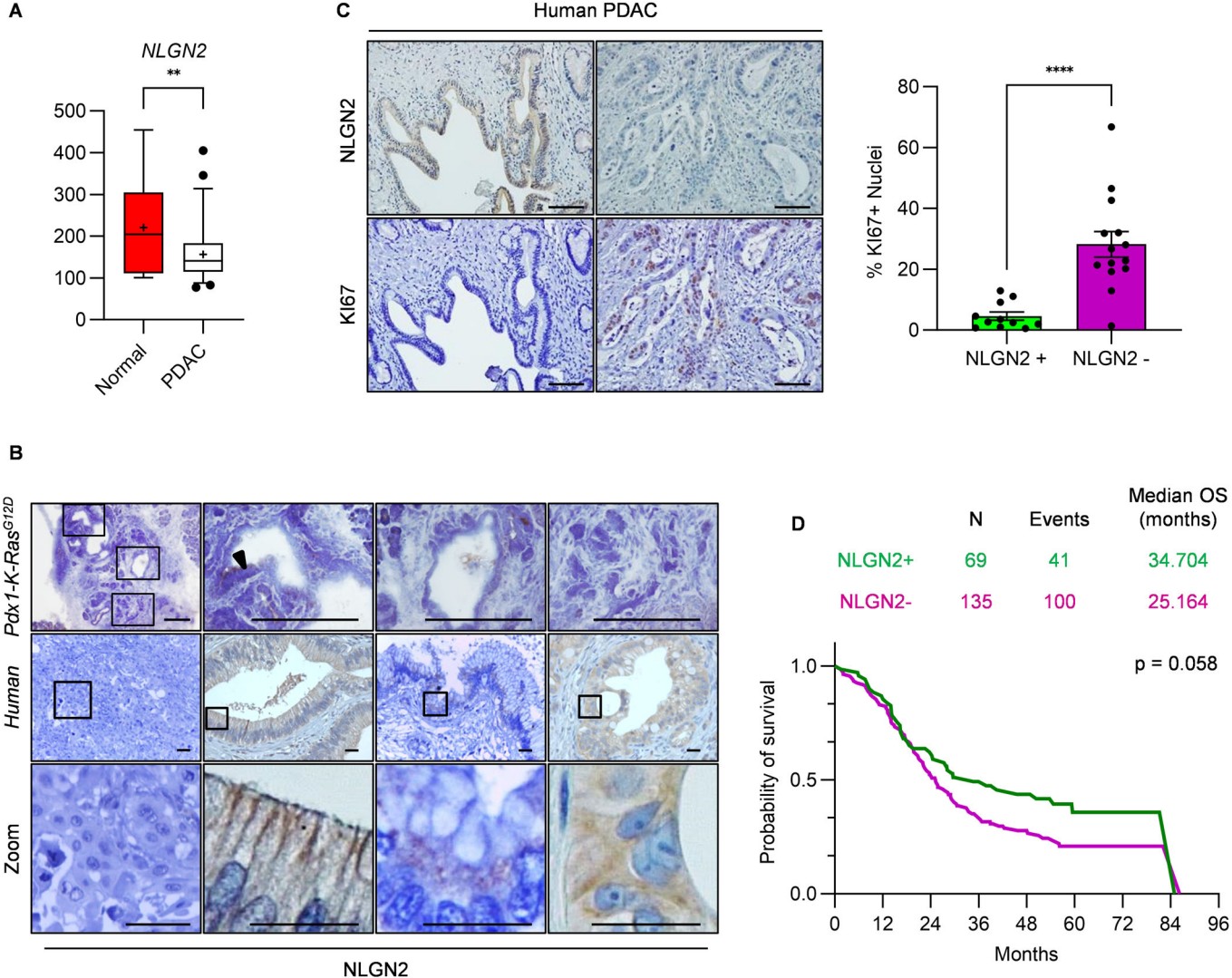

**Figure 2. NLGN2 expression in PDAC is heterogeneous and correlates with malignancy.**

(A) Quantitative analysis using the MERAV online database demonstrated a significant reduction in *NLGN2* mRNA expression in PDAC tissues compared to normal pancreatic tissues. (B) Top: Immunohistochemical analysis of NLGN2 expression in PDAC tumors of *Pdx1-K-Ras^G12D* mice. From left to right: the gross appearance of mouse PDAC tumors. Square boxes identify tissue areas shown at higher magnification; PDAC areas showing NLGN2-positive cells (arrowheads); PDAC glands without expression of NLGN2; areas with metaplastic ducts and inflammatory infiltrate negative for NLGN2. Middle: different NLGN2 distribution patterns found in human PDAC. From left to right: while most tumors were NLGN2-negative (first panel), positive tumors showed apical membranous (second) or intracellular (third) NLGN2 localization or unpolarized diffuse NLGN2 localization (fourth). The square boxes identify tissue areas shown at higher magnification in the bottom panels (Zoom). (C) NLGN2 expression in PDAC negatively correlates with the percentage of Ki67+ nuclei, as shown by immunohistochemical analysis (left) and quantification (right). The graphs report the Ki67+ nuclei percentage. (D) Kaplan–Meier survival curves for NLGN2-positive and NLGN2-negative PDAC patients. The panel indicates the number of events (deaths) and the median survival time of NLGN2-positive and NLGN2-negative patients. Data information: in (A) the center lines show the medians and the crosses indicate sample means; the box limits indicate the 25th and 75th percentiles; the whiskers extend from the 5th to 95th percentile, outliers are represented by dots; unpaired *t*-test: **$p < 0.01$; $n = 57$ PDAC, 16 normal pancreas samples. In (B) mouse PDAC tumors scale bar: 0.2 mm, the images are representative of ten mice; human tumor scale bar: 0.05 mm. (C) Left: scale bar: 0.1 mm; right: data were presented as mean ± SEM (NLGN2-negative samples, $n = 14$; NLGN2-positive samples, $n = 11$; unpaired *t*-test: ****$p < 0.0001$). (D) OS was analyzed by the Kaplan–Meier method and compared using the log-rank test ($p = 0.058$). Source data are available online for this figure.

expression was associated with worse survival, although the correlation was not statistically significant (hazard ratio 1.47, 95% CI 0.99–2.19, $p = 0.058$) (Table 2).

Analysis based on the *NLGN2* transcript level in 168 PDAC samples available in a dataset from The Cancer Genome Atlas (TCGA) corroborated the above analysis. Interestingly, the subgroup of patients with higher *NLGN2* expression showed better overall survival (OS) ($p < 0.05$) than the subgroup exhibiting lower *NLGN2* expression (Fig. EV3B).

## Nlgn2 regulates contact inhibition

To understand how NLGN2 expression is linked with the changes in proliferation and polarity during the progression from low- to

**Table 1. Main clinical characteristics of patients with PDAC included in survival analysis (n = 204).**

| | Positive NLGN2 expression (69 patients) N of pts (row percentages) | Negative NLGN2 expression (135 patients) | P value |
|---|---|---|---|
| Clinical stage | | | 0.28 |
| *Missing info* | *1* | *–* | |
| I–II | 62 (34.8%) | 116 (65.2%) | |
| III–IV | 6 (24.0%) | 19 (76.0%) | |
| Surgical margins | | | 0.47 |
| Negative | 46 (35.7%) | 83 (64.3%) | |
| Positive | 23 (30.7%) | 52 (69.3%) | |
| Perioperative chemotherapy | | | 0.51 |
| *Missing info* | *6* | *18* | |
| No | 13 (31.0%) | 29 (69.0%) | |

**Table 2. Multivariable analysis for overall survival.**

| Covariate | Features | Hazard ratio (95% CI) | P value |
|---|---|---|---|
| NLGN2 | Negative versus positive | 1.47 (0.99–2.19) | 0.058 |
| Margin | Positive versus negative | 1.55 (1.06–2.27) | 0.024 |
| Stage | III–IV versus I–II | 0.96 (0.57–1.63) | 0.88 |
| (Neo)adjuvant regimen | Yes versus No | 0.51 (0.33–0.78) | 0.002 |

**Table 3. mRNA expression of Neuroligin family genes in WT-HPDE[a].**

| | *NLGN1* | *NLGN2* | *NLGN3* | *NLGN4X* | *NLGN4Y* | TBP |
|---|---|---|---|---|---|---|
| WT-HPDE | N.D.[b] | 26.97 | N.D. | N.D. | N.D. | 24.50 |
| HUVEC | 27.33 | 26.72 | 37.13 | 31.25 | 36.72 | 25.70 |

[a]Real-time PCR of neuroligin genes and *TBP* as housekeeping gene in WT-HPDE and human endothelial cells (HUVEC). The data shown are representative of three individual experiments done on three different cell cultures.
[b]Raw Ct indicating *NLGNs* and *TBP* absolute expression levels. N.D. stands for "non detected".

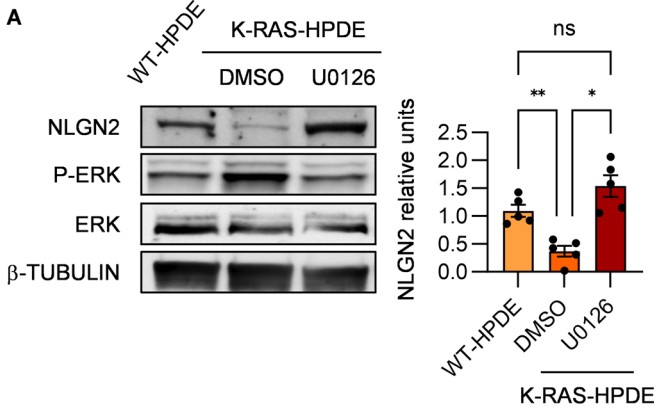

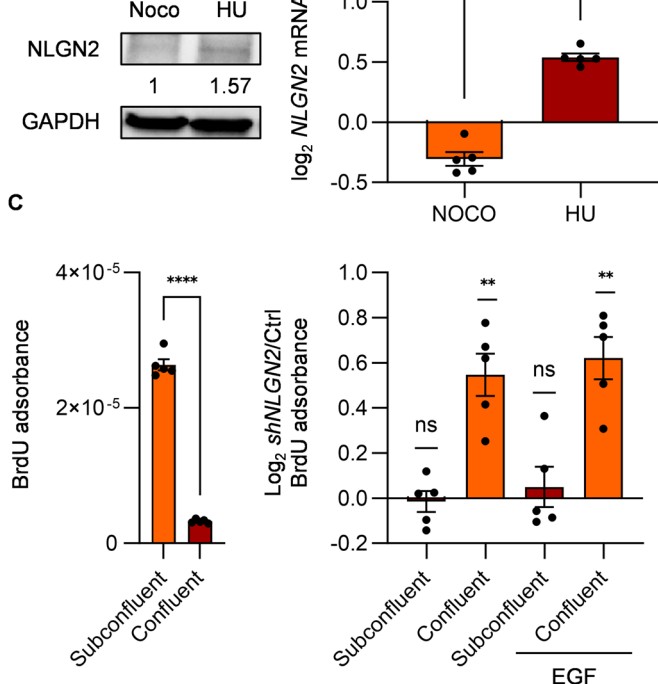

high-grade PanIN, given that acinar cells transdifferentiate into ductal-like precursors before PanIN transformation (ADM) (Guerra et al, 2007; Shi et al, 2013), we used the immortalized human pancreatic ductal epithelial cell line WT-HPDE for easier in vitro handling. In these cells, *NLGN2* was the only *neuroligin* family gene expressed (Table 3). The expression of constitutively active K-RAS[G12V] (generating K-RAS-HPDE cells) greatly reduced the expression of NLGN2, and the treatment of K-RAS-HPDE cells with the MEK inhibitor U0126 (Yip-Schneider and Schmidt, 2003) rescued NLGN2 expression (Fig. 3A). Since low-grade PanINs have a low rate of cell proliferation, express NLGN2 and mutated K-RAS, the K-RAS-dependent NLGN2 downregulation is likely

blocked by events which impairs K-RAS-dependent cell proliferation, such as oncogene-induced senescence. To verify that NLGN2 expression depends on cell cycle progression, we monitored the expression of NLGN2 during the cell cycle in WT-HPDE by treating cells with hydroxyurea (HU) and nocodazole (Noco) to synchronize them in the G1/S and G2/M phases, respectively (Fig. EV3C,D). We observed that the NLGN2 level oscillates during cell proliferation, as it is significantly reduced in cells at the end of S phase compared to G1 phase (Fig. 3B). To exclude a direct effect of HU or Noco on *NLGN2* expression, *NLGN2* modulation was replicated with superimposable results by treating the cells with thymidine or colcemid, which synchronize cells in the G1/S and G2/M phases, respectively. We then sought to determine whether NLGN2 modulation is just a secondary consequence of cell proliferation or might also play a role in its regulation. Notably, as in cells carrying nontargeting *shRNA* (Ctrl), NLGN2 silencing in subconfluent WT-HPDE cells (*shNLGN2*) did not lead to

Figure 3.  NLGN2 regulates cell contact inhibition.

(A) NLGN2 expression is modulated in subconfluent WT-HPDE and K-RAS-HPDE cells. Western blot analysis (left) of the indicated proteins in WT-HPDE and K-RAS-HPDE cells treated or untreated with U0126 (10 μM) for 8 h. Right: the graphs show the results of densitometric analysis of NLGN2 expression normalized to β-tubulin expression. (B) Left: Western blot analysis of NLGN2 in nocodazole (Noco)- and hydroxyurea (HU)-treated WT-HPDE cells; relative densitometric analysis of NLGN2 expression normalized to GAPDH expression is indicated. Right: fold change in NLGN2 mRNA expression determined by real-time PCR in synchronized HU-treated cells compared with Noco-treated WT-HPDE cells. (C) Left: BrdU absorbance normalized to the cell number in Ctrl WT-HPDE cells, showing reduced BrdU incorporation in confluent cells compared to subconfluent cells. Right: BrdU absorbance in shNLGN2 WT-HPDE normalized on Ctrl, untreated or treated with 20 ng/ml EGF for 8 h. Data information: in (A) data were presented as mean densitometry ± SEM normalized on the mean, n of biological replicates = 5 (one-way ANOVA: *p < 0.05; **p < 0.01; non significant: ns). In (B) data were presented as mean log$_2$ mRNA change ± SEM normalized on untreated cells of technical triplicates, biological replicates = 5 (paired t-test: ***p < 0.001). (C) Left: normalized mean ± SEM BrdU absorbance of technical triplicates, biological replicates = 5 (paired t-test: ****p < 0.0001); right: mean log$_2$ BrdU adsorbance ± SEM of technical triplicates, biological replicates = 5 (one sample t-test: **p < 0.01; non significant: ns). Source data are available online for this figure.

significant alterations in cell proliferation, as assessed by BrdU incorporation (Fig. 3C). However, we also analyzed the consequence of NLGN2 silencing in confluent cells, given that proliferation can be regulated by contact inhibition, which in turn depends on polarity complex integrity (Vaccari and Bilder, 2009). First, we demonstrated the contact inhibition ability of WT-HPDE cells, as evidenced by the reduction in BrdU incorporation in confluent cultures compared to subconfluent cultures (Fig. 3C). Interestingly, NLGN2 expression increased greatly as WT-HPDE cells reached confluence (Fig. 4A), consistent with the observation that NLGN2 expression was increased in G1/S-phase cells. In contrast to the observation in subconfluent cells, NLGN2 silencing in confluent cells reduced contact inhibition, as shown by significantly increased BrdU incorporation (Fig. 3C).

We further demonstrated that NLGN2 modulates the sensitivity of WT-HPDE cells to EGF, which plays a central role in PDAC progression (Navas et al, 2012). EGF stimulation resulted in higher BrdU incorporation in shNLGN2 WT-HPDE, even in confluent cultures. Notably, EGF stimulation did not result in differences in proliferation between Ctrl and shNLGN2 subconfluent cells (Fig. 3C).

## NLGN2 regulates cell polarity

To provide visual proof of the effect of NLGN2 modulation on cell morphology, we imaged WT-HPDE cells at different culture time points during growth from a subconfluent to a confluent state (Fig. 4A). After Ctrl WT-HPDE cells reached "early" confluence, they were not totally contact-inhibited and proliferated further, albeit at a decreased rate, as shown by the reduced BrdU incorporation (Fig. 3C). Ctrl cells proliferated with regard to adjacent cells maintaining a monolayer structure, and each cell decreased in size to increase the available space for new cells (Fig. 4A,B). We call this state of a crowded cell monolayer "mature confluence" to distinguish it from "early confluence", with larger confluent cells. Importantly, shNLGN2 WT-HPDE cells not only showed greater proliferation at confluence but also formed clusters

of crowded and overlapping cells growing on multiple layers (Fig. 4A,B). This phenotype underscores that NLGN2 loss might affect not only cell proliferation but also mechanisms regulating aspects of cellular organization in space, such as cell polarity.

To further understand the impact of NLGN2 silencing on cell polarity, we used an in vitro 3D cystogenesis assay. In contrast to well-established cell models used for cyst polarity studies (Jaffe et al, 2008; Schoenenberger et al, 1991), WT-HPDE cells did not form perfect cysts with a layer of adjacent single cells, despite several attempts to adjust the culture conditions. However, WT-HPDE promptly formed cysts characterized by a hollow lumen, which requires the cell to be able to distinguish between its apical and basolateral domains (Jaffe et al, 2008). Conversely, shNLGN2 WT-HPDE cells formed cysts with dimensions similar to those of cysts formed by Ctrl cells but filled or composed of thick cell layers (Fig. 4C). Moreover, we sought to determine whether NLGN2-mediated regulation of contact inhibition also impacts cystogenesis. After 3 weeks of culture, Ctrl cysts displayed growth arrest, but the proliferation of shNLGN2 cysts continued, as shown by the increased EdU incorporation (Fig. 4D). To ensure that the above 2D and 3D phenotype in shNLGN2 cells are not due to aspecific targets of the exogenous shRNA, we validated the reproducibility of this experiments using 5 shRNA targeting NLGN2 sequence in different regions with superimposable results (Fig. EV4A,B).

To confirm the role of NLGN2 in regulating polarized cystogenesis, we generated shRNA-insensitive full-length NLGN2 construct with a MYC tag to track its localization (NLGN2-MYC), and expressed it in shNLGN2 WT-HPDE cells (generating shNLGN2 + NLGN2-MYC cells) (Fig. EV4C). These cells were stable in culture for many passages without showing gross morphological or growth differences compared with Ctrl and shNLGN2 cells at subconfluence, and they were arranged in a stable cell monolayer at early confluence. However, when we allowed them to stay at a confluence, the cell layer detached from the culture well (Fig. EV4D) before reaching mature confluence. We observed no such phenomenon with either Ctrl or shNLGN2 cells, which typically can stay at confluence for an undetermined number of days. We hypothesize that this effect is due to the higher NLGN2-MYC expression and activity in shNLGN2 + NLGN2-MYC cells than in Ctrl cells (Fig. EV4C), which sensitize them to confluence and induce contact inhibition strongly enough that the cell culture becomes unstable.

Because WT-HPDE cells did not seem to be the best cell system in which to study polarity aspects of cystogenesis and poorly tolerated NLGN2-MYC overexpression, we used CACO2 cells, which are a well-established model of epithelial polarization (Jaffe et al, 2008). Upon NLGN2 silencing, CACO2 cysts also acquired aberrant features, such as intraluminal cell growth and lumen absence (Fig. 4E). Immunofluorescence analysis showed that shNLGN2 CACO2 cysts exhibited unpolarized distribution of ZO1 and ECAD, which were localized at apical and basolateral sites, respectively, in Ctrl CACO2 cysts (Fig. 4F). MYC staining in shNLGN2 CACO2 + NLGN2-MYC cells showed MYC protein localization in the cell membrane in 2D culture and at the apical domain in 3D culture, similar to its distribution in mouse pancreata (Fig. EV4E). Strikingly, expression of NLGN2-MYC in shNLGN2 CACO2 cells rescued the ability to produce normal cysts characterized by a single-cell layer, a hollow central lumen, and polarized distributions of ECAD and ZO1 (Fig. 4E,F).

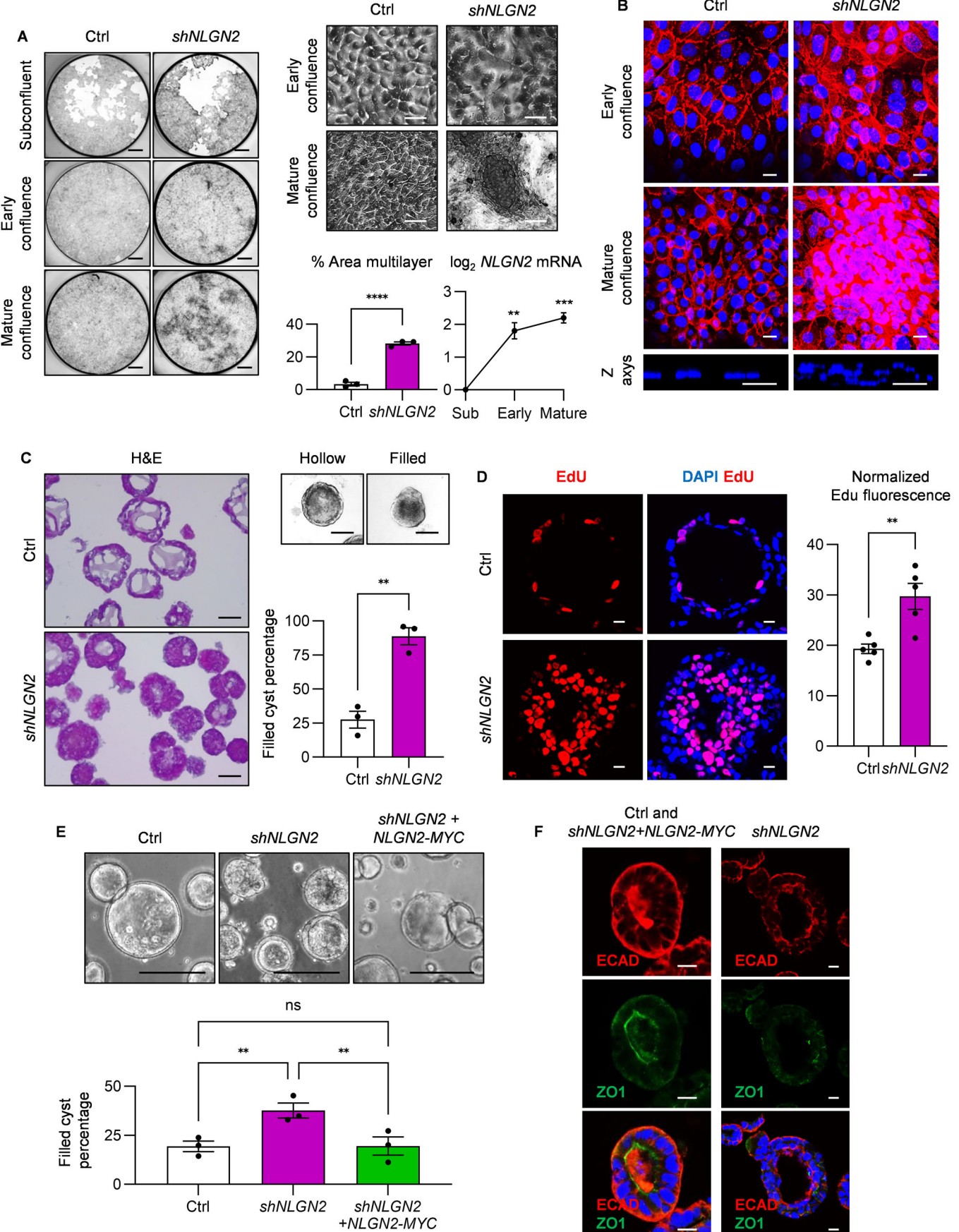

**Figure 4. NLGN2 regulates cystogenesis in three-dimensional in vitro culture.**

(A) Left: Fixed cultures of Ctrl and *shNLGN2* WT-HPDE cells stained with crystal violet, indicating overgrowing *shNLGN2* cells at the mature confluence. Right: Optical micrographs of confluent cultured Ctrl and *shNLGN2* WT-HPDE cells. The difference between the cobblestone appearance of Ctrl WT-HPDE cells and the overlapping morphology of *shNLGN2* WT-HPDE cells can be appreciated (top). Bottom left: percentage of the culture area at mature confluence occupied by multilayer cells in Ctrl and *shNLGN2* WT-HPDE. Bottom right: $\log_2$ *NLGN2* mRNA expression levels in cells at early and mature confluence, normalized to those in subconfluent cells. (B) Top: immunofluorescence staining of Ctrl and *shNLGN2* WT-HPDE cells at early and mature confluence. Mature *shNLGN2* image consists of a maximum projection of confocal stacks, to represent cells growing on multiple layers. E-Cadherin (red) was stained to highlight cell contacts and morphology. Bottom: z-axis images of the cultured cells, showing DAPI-stained nuclei (blue) and overlapping cells in the *shNLGN2* WT-HPDE cell culture compared to the Ctrl cell culture. (C) Left: Sectional images of 3D cultured WT-HPDE cells stained with hematoxylin and eosin, showing cysts with hollow lumens in Ctrl samples (top) and aberrant cysts after *NLGN2* silencing (bottom). Aberrations are manifested as cysts without a central lumen or as a prominent overlapping cell layer. Right: Cyst morphology quantification indicating the prevalence of filled cysts in *shNLGN2* cells. To assess cell morphology, we used optical images of whole cysts (top) to avoid artifacts induced by section processing. (D) Left: incorporation of the proliferation marker EdU in 3-week-cultured WT-HPDE cysts, showing increased proliferation of *shNLGN2* cells compared to Ctrl cells. Right: EdU fluorescence quantification normalized to the number of nuclei. (E) Top: confocal images of 3D cultured CACO2 cells showing monolayer cysts with hollow lumens in Ctrl samples and aberrant cysts after *NLGN2* silencing. Aberrations are manifested as cysts without a central lumen and with multilayered cells. Re-expression of NLGN2-MYC in *shNLGN2* CACO2 cells restored the cyst phenotype. Bottom: the graph report filled cyst percentage quantification. (F) ECAD (red) and ZO1 (green) localization in Ctrl and *shNLGN2* CACO2 + NLGN2-MYC cells (left) and in *shNLGN2* CACO2 cells (right) grown in 3D cultures. Data information: in (A) left: scale bar: 1mm; right scale bar: 25 µm; bottom left: data were presented as mean ± SEM of biological replicates = 3 (Unpaired *t*-test: ****$p < 0.0001$); bottom right: the graph report the mean mRNA $\log_2$ change ± SEM of technical triplicates, biological replicates = 5 (one sample *t*-test: **$p < 0.01$; ***$p < 0.001$). (B) Scale bar: 10 µm. (C) Scale bar: 0.1 mm; the graph report the mean filled cyst percentage ± SEM, biological replicates = 3 (cyst Ctrl: $n = 242$; cyst *shNLGN2*: $n = 367$; unpaired *t*-test: **$p < 0.01$). (D) Left scale bar: 10 µm; right: the graph shows the mean normalized fluorescence ± SEM in 14 representative cysts for each sample, biological replicates = 5 (unpaired *t*-test: **$p < 0. 01$). (E) Top scale bar: 0.1 mm; bottom: the graph report the mean filled cyst percentage ± SEM, biological replicates = 3 (cyst Ctrl: $n = 804$; cyst *shNLGN2*: $n = 876$; cyst *shNLGN2* + NLGN2-MYC: $n = 867$; one-way ANOVA: **$p < 0.01$; non significant: ns). (F) Scale bar: 10 µm. Source data are available online for this figure.

## NLGN2 regulates YAP activation through PALS1/PATJ

We next sought to determine how NLGN2 connects cell polarity and contact inhibition. To this end, we considered that YAP regulation is a valid candidate due to its proven role in the orchestration of such processes (Aragona et al, 2013; Gumbiner and Kim, 2014). Interestingly, it has been reported that NLGN2 binds to PATJ, a component of the Crumbs polarity complex, which senses and regulates cell polarity and leads to contact inhibition through the Hippo pathway. In mature confluent *shNLGN2* WT-HPDE cells, we observed overall downregulation of both PATJ and PALS1 (Fig. 5A), with a strong decrease in PATJ expression, in accordance with previous indications that a reduction in their interaction could increase their susceptibility to degradation (Straight et al, 2004). Consistently, NLGN2 coimmunoprecipitated with PATJ (Fig. 5B), and NLGN2 silencing reduced the formation of the PATJ/PALS1 complex (Fig. 5C). We therefore checked whether the reduced formation of the PALS1/PATJ complex affects YAP activation through recruitment of the adapter protein AMOT. In mature confluent WT-HPDE cells, we observed the formation of the AMOT-YAP complex independent of NLGN2 expression (Fig. 5D), in agreement with previous studies reporting that this complex is formed independent of AMOT and YAP phosphorylation (Moleirinho et al, 2017). AMOT-YAP coimmunoprecipitated with PALS1 and PATJ in mature confluent WT-HPDE cells, and NLGN2 silencing greatly reduced the formation of this complex (Fig. 5E).

The reduction in YAP recruitment to PALS1/PATJ in mature confluent *shNLGN2* WT-HPDE cells resulted in a reduction in phosphorylated YAP-S127 compared to that in Ctrl cells (Fig. 5F). To monitor the effect of NLGN2 expression on YAP activation, we measured the mRNA levels of the YAP target genes CTGF, CYR61, AREG, and HBEGF (Piccolo et al, 2014; Rozengurt et al, 2018) in Ctrl cells, showing that the mRNA expression of these genes is reduced as Ctrl cells progress from a subconfluent to a confluent state (Fig. 5G). However, in *shNLGN2* cells, YAP target gene expression at confluence was significantly increased compared to that in Ctrl cells (Fig. 5G).

These findings were consistent with the YAP expression pattern in *Elast-K-Ras^{G12V}* and *Pdx1-K-Ras^{G12D}* mice. While normal acinar cells were YAP-negative, we observed the presence of nuclear YAP in ADM (Figs. 5H and EV5A), consistent with previous findings (Gruber et al, 2016; Zhang et al, 2014). However, as ADM progressed to low-grade PanIN, YAP nuclear expression was significantly reduced, consistent with the halted proliferation of these lesions. Notably, while low-grade PanINs in *Elast-K-Ras^{G12V}* mice were completely devoid of YAP expression, low-grade PanINs in *Pdx1-K-Ras^{G12D}* mice expressed YAP, although it was localized mostly in the cytoplasm. Finally, YAP was abundantly localized in the nucleus in high-grade PanINs in both GEMMs (Figs. 5 and EV5A). In *Pdx1-K-Ras^{G12D}* mice, tumors were surrounded by a range of low- and high-grade glands, often with different grades of dysplasia within the same lesion. Very intriguing, nuclear YAP expression was heterogeneous but still highly correlated with gland polarization (Fig. EV5B). We finally asked whether YAP activation is responsible for the phenotypes observed in the absence of NLGN2. In mature confluent *shNLGN2* WT-HPDE, characterized by a heterogeneous pattern of monolayer and multilayer cells (Figs. 5I and EV5C), we observed that the monolayer displayed membranous PATJ expression and low proliferation, while overgrowing cells lost PATJ expression and exhibited high proliferation rate (Fig. 5I). YAP silencing in mature confluent *shNLGN2* cells (Fig. EV5C) induced a significant reduction of the protruding multilayers, which were found as floating rafts in the culture medium in 24 h (Fig. 5J). When these rafts were transferred to new culture dishes, they were not able to regrow either in adhesion or in suspension and rapidly fragmented, indicating that a cell death program was ongoing (Fig. 5J). These effects were replicated using two different *shRNA* for YAP, with superimposable results. Strikingly, mature confluent *shNLGN2* organized as a monolayer maintained confluence and were unaffected by YAP silencing.

## Discussion

Here, we demonstrate a novel function for NLGN2 in pancreatic exocrine cell biology and its importance in regulating the low- to high-grade PanIN switch, cell proliferation, and cell polarity through a YAP-mediated mechanism.

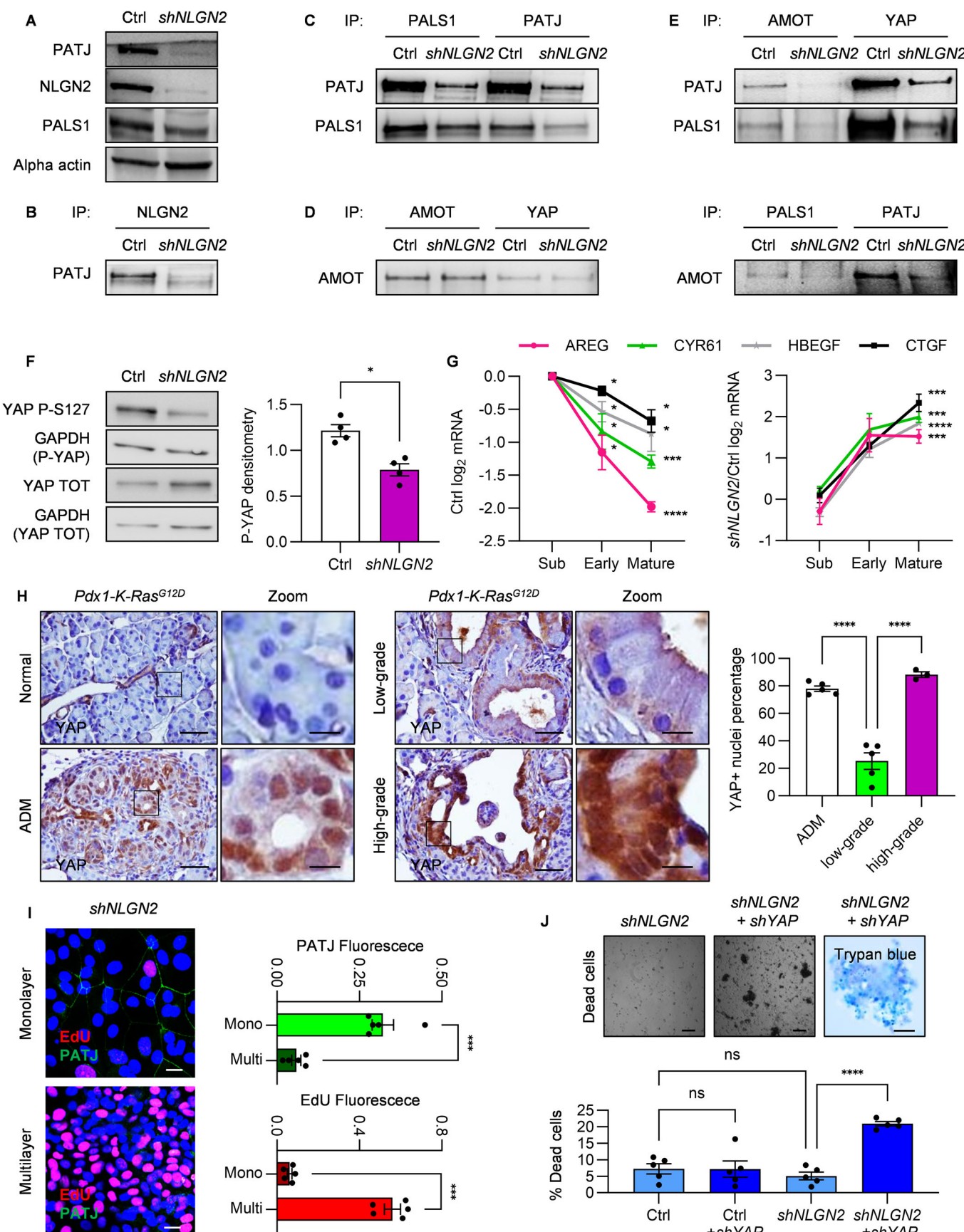

**Figure 5.   NLGN2 regulates YAP through PALS1/PATJ complex formation.**

(A) Western blot analysis in Ctrl and *shNLGN2* WT-HPDE cells at mature confluence, indicating the reduced expression of PATJ and PALS1 in *shNLGN2* cells. (B) Immunoprecipitates from cell lysates (IP) of Ctrl and *shNLGN2* WT-HPDE cells at mature confluence. The western blot shows PATJ and NLGN2 interaction. (C) Immunoprecipitates from cell lysates (IP) of Ctrl and *shNLGN2* WT-HPDE cells at mature confluence. The western blot of the indicated proteins shows reduced interaction of PALS1 and PATJ in *shNLGN2* cells. (D) Immunoprecipitates from cell lysates (IP) of Ctrl and *shNLGN2* WT-HPDE cells at mature confluence. The western blot of AMOT shows its interaction with YAP independently from the expression of NLGN2. (E) Immunoprecipitates from cell lysates (IP) of Ctrl and *shNLGN2* WT-HPDE cells at mature confluence. The western blot of the indicated proteins shows reduced interaction of AMOT and YAP with PALS1 and PATJ in *shNLGN2* cells. (F) Western blot analysis of total and phospho-S127 (P-S127) YAP in Ctrl and *shNLGN2* WT-HPDE cells at mature confluence. Total and phosphorylated proteins were decorated on different membranes, and GAPDH was used to normalize the level of P-S127 YAP to that of total YAP. The graph report the densitometry quantification. (G) Real-time PCR analysis of *AREG*, *HBEGF*, *CTGF*, and *CYR61* expression in subconfluent, early confluent and mature confluent cells. The left panel shows $\log_2$ gene expression values normalized to those in subconfluent Ctrl cells. The right panel shows relative gene expression in *shNLGN2* cells compared to Ctrl cells. (H) Immunohistochemical analysis of total YAP expression in normal pancreatic, ADM, and low- and high-grade PanIN tissues of 7-month-old *Pdx1-K-Ras*$^{G12D}$ mice. The right panels show higher-magnification versions of the corresponding images (Zoom). The graph shows the percentage of YAP-positive nuclei. (I) Left: EdU and PATJ immunofluorescence staining of mature confluent *shNLGN2* WT-HPDE cells, showing heterogeneous staining between monolayer and multilayer cells. The right graphs shows fluorescence quantification normalized on cell nuclei. (J) Top: optical images of detached cells in the medium of mature confluent *shNLGN2* and *shNLGN2*+*shYAP*, showing single dead cells from *shNLGN2* culture and rafts from multilayer cells from *shNLGN2*+*shYAP* WT-HPDE. Right panel shows trypan blue staining of cell rafts from *shNLGN2*+*shYAP* cells. The cells in the medium were transferred to a new dish to take the photo, in order to avoid the interference of the underlying monolayer cells. Bottom: the graph shows cell death percentage in mature confluent *shNLGN2* and *shNLGN2*+*shYAP* measured by trypan blue staining, Ctrl cells do not show increased cell death compared to *shNLGN2* cells, and *shYAP* did not induce cell toxicity in Ctrl cells. Data information: in (A–E) the images are representative of three biological replicates with similar results. In (F) the graph report the mean ± SEM normalized on the mean, biological replicates = 4 (paired *t*-test: *$p < 0.05$). In (G) the graphs report the $\log_2$ mRNA mean fold change ± SEM of technical triplicates, biological replicates = 5 (one sample *t*-test: *$p < 0.05$; ***$p < 0.001$; ****$p < 0.0001$). For visual clarity, only mature confluence significance is reported in the *shNLGN2*/Ctrl graph, early confluence significance is $p < 0.05$ for AREG, CYR61, and HBEGF; $p < 0.0001$ for CTGF; Subconfluence difference is not significant. (H) Scale bar: 50 µm; Zoom scale bar: 10 µm; the graph reports nuclei percentage mean ± SEM (*n* mice with ADM and low-grade PanIN = 5; *n* mice with high-grade PanIN = 3; nuclei: ADM: *n* = 2627; low-grade PanIN: *n* = 3022; high-grade PanIN: *n* = 2430; one-way ANOVA: ****$p < 0.0001$). (I) Left: scale bar: 25 µm. Right: the graphs shows mean fluorescence ± SEM, biological replicates = 5 (paired *t*-test: ***$p < 0.001$). (J) Top left and mid: scale bar: 150 µm, right trypan blue photo scale bar: 50 µm; bottom mean trypan blue positive cells percentage ± SEM, biological replicates = 5 (one-way ANOVA: ****$p < 0.0001$; non significant: ns). Source data are available online for this figure.

We found that a set of neural genes, particularly specialized in synaptic structural stability and function, is highly expressed in pancreatic acinar cells and lost in high-grade PanIN. Intuitively, acinar cells adopt molecular machinery for digestive enzyme secretion that is similar to that necessary for neurotransmitter release; however, our study found that these proteins might cover new functions. Our analysis focused on NLGN2, which is segregated at the apical domain of normal acinar cells, and that its expression patterns progressively change during the progression from low- to high-grade PanIN in the human pancreas and in *Elast-K-Ras*$^{G12V}$ and *Pdx1-K-Ras*$^{G12D}$ mouse models. In low-grade PanINs, which are not characterized by evident changes in cell polarity, NLGN2 remains expressed, albeit in a diffuse cytosolic pattern, and its expression is lost in high-grade PanINs.

Similarly, pancreatitis in caerulein-treated mice resulted in the mobilization of NLGN2 from the apical domain to the cytosol in acinar cells and ADM, and this relocalization was reversed after inflammatory injury cheased. This phenomenon could explain the heterogeneous pattern of NLGN2 localization in the normal human pancreas as being derived from the history of pancreatitis, therapeutic treatments, and individual exposome.

Several papers have underscored the relevance of YAP during the initial stages of K-RAS-induced ADM transformation and in established PDAC (Gruber et al, 2016; Zhang et al, 2014). However, data pointing to a role for YAP in PanIN progression, more specifically, with regards to discriminating between the growth arrest observed in low-grade PanINs and later overcome in high-grade PanINs are scarce. Mutated K-RAS supports ADM, which would otherwise be reversible (Grimont et al, 2022), and we speculate that it also supports delocalization of NLGN2, which may play a partial role in the loss of polarity, and induces YAP nuclear translocation, which initiates or at least favors the progression from ADM to low-grade PanIN. Then, the oncogene-induced senescence program negatively interferes with cell cycle progression, resulting

in higher NLGN2 expression due to a blockade in the G1 cell cycle phase. Additionally, proliferation might be blocked by activation of the Hippo pathway, which can be triggered due to its size sensing of the newly growing PanIN gland (Aragona et al, 2013). Further variations in the mutational landscape and the increased availability of growth factors in the inflammatory microenvironment (Gruber et al, 2016) might reduce NLGN2 expression and trigger YAP overactivation. Intriguingly, loss of the WT-*K-RAS* allele can promote PDAC malignancy through YAP overactivation (Yan et al, 2021). This is likely to occur in the later stage of PanIN and may promote NLGN2 downregulation through sustained MAPK pathway signaling.

This proposed model is supported by our data demonstrating a high level of nuclear YAP in ADM, which decreased dramatically in low-grade PanIN and increased again in high-grade PanIN, concomitant with the increased proliferation and loss of apicobasal polarity.

We can hypothesize that YAP-dependent overexpression of autocrine factors involved in PDAC progression, such as AREG and HBEGF (Wagner et al, 2002; Wang et al, 2016; Yotsumoto et al, 2010), plays a role in this process, consistent with studies showing that PanINs progression requires EGFR signaling (Navas et al, 2012).

While the role of YAP in PDAC is widely documented, little is known about the role of the Crumbs complex and about polarity-regulating mechanisms in general in PDAC onset. From a mechanistic perspective, our in vitro data suggest two key results, which define cell behavior at confluence. First, NLGN2 is necessary for the formation of the PALS1/PATJ complex in confluent cells. We observed a general decrease in PALS1 and PATJ proteins, and whether this resulted from the reduction in PALS1 and PATJ expression or whether the affinity between these proteins is reduced in *shNLGN2* cells, leading to their degradation, is unclear. NLGN2 is known mostly for its role as a transmembrane adherence protein in synapses, where it regulates CDC42 Guanine Nucleotide

Exchange Factor 9 (ARHGEF9) and is responsible for the formation of the Gephyrin submembranous lattice (Poulopoulos et al, 2009; Soykan et al, 2014). These events allow the correct localization of neurotransmitter receptors and scaffold proteins, as well as their interactions with the cytoskeleton. Importantly, signaling orchestrated by polarity regulators must occur specifically in the proper subcellular compartment (Zihni and Terry, 2015), as recently shown by superresolution imaging of PALS1/PATJ in proximity to tight junctions (Mangeol et al, 2022). Notably, the PALS1/PATJ complex interacts with the transmembrane polarity determinant CRB; however, only PALS1 directly interacts with CRB (Roh et al, 2002). Therefore, the promotion of PALS1/PATJ complex formation might be regulated by NLGN2 through the correct transport or localization of PATJ to specific subcellular domains, where it can interact with PALS1. Additionally, CDC42 regulates CRB localization (Harris and Tepass, 2008) and NLGN2 might also play a role in this process through ARHGEF9.

Second, YAP is inhibited in confluent cells. Given that NLGN2 silencing has no apparent effect on proliferation at subconfluence, it is likely that a sensor of cell crowding connected with actin cytoskeleton rearrangement (Aragona et al, 2013; Dupont et al, 2011) signals to recruit AMOT-YAP to the PALS1/PATJ complex at the plasma membrane of confluent cells (Moleirinho et al, 2017). This is strongly suggested by the heterogeneous pattern of proliferation and PATJ expression that we observed in mature confluent *shNLGN2* cells, which indicates that NLGN2 absence alone is not sufficient to induce loss of polarity and contact inhibition: a factor like mechanic tension, which might be distributed not-uniformly in the cell monolayer could lead to YAP activation, which in normal condition is inhibited by NLGN2. This phenomenon would be consistent with the growth arrest observed in low-grade PanINs, which exhibit cell crowding and pseudostratification, as well as with the ability of WT-HPDE cysts to grow for two weeks before growth cessation. In this context, the role of NLGN2-dependent YAP recruitment to the PALS1/PATJ complex might not be limited only to the formation of this complex but may also extend to actin cytoskeleton rearrangement (Bazellieres et al, 2018), which later acts as a cell crowding sensor and initiates the sequence of events leading to YAP inhibition. Overall, the loss of contact inhibition in *shNLGN2* cells, which promotes the formation of proliferating multilayered 2D cultures and aberrant cysts, is reminiscent of the high-grade PanIN architecture and of the phenotype observed in the MDCK cell line carrying mutated *K-Ras* (Schoenenberger et al, 1991).

To investigate the clinical significance of the decrease in NGLN2 expression in PDAC, we performed Kaplan–Meier survival analysis. The results, even if characterized by the limited statistical power of this analysis, indicated that loss of NLGN2 expression was associated with worse survival. However, a recent bioinformatic analysis performed on public datasets indicated that the combination of *NLGN2* with *CALB2*, *NCAPG*, and *SERTAD2* represents a signature with significant prognostic value in PDAC (Zhang et al, 2022). Similarly, NLGN2 was reported to be a prognostic marker in breast cancer (Zhang et al, 2021). Therefore, it is worth exploring whether NLGN2 is a potentially useful target in PDAC management. Since PDAC has a very poor prognosis even when diagnosed early, more efforts to improve the differential diagnosis of low- to high-grade PanINs are needed through biomarker development. In this regard, it could be interesting to analyze the genes with

expression modulation as a consequence of NLGN2 reduction or to evaluate NLGN2 as a biomarker, given recent findings demonstrating the possibility of detecting NLGN2 in plasma (Thavarajah et al, 2020). Furthermore, studying the mechanisms underlying the low- to high-grade PanIN transition might provide innovative therapeutic opportunities to delay or reverse PanIN progression, which could include strategies based on restoring NLGN2-mediated signaling. This idea is supported by the results of studies focusing on the role of NLGN2 in insulin secretion and on the potential of NLGN2 mimetics as tools for treating patients with diabetes (Munder et al, 2017; Munder et al, 2019).

Finally, we attempted to model the transition from low- to high-grade PanIN via NLGN2 silencing. Although a limit of this study is the lack of mouse genetics models able to demonstrate the causative connection between NLGN2 and YAP in PanIN progression, our results provide unprecedented insights into the role played by loss of polarity and contact inhibition in early PDAC stages and indicate the axis NLGN2/YAP as a promising target for new strategies useful in PDAC clinical management.

## Methods

### Mouse breeding and treatment

*Pdx1-Cre;K-Ras$^{+/LSLG12D}$* (*Pdx1-K-Ras$^{G12D}$*) (Hingorani et al, 2003) mice were generated in the Animal Facility of the Molecular Biology Center of the University of Torino (Torino, Italy) (Cappello et al, 2013). *Elas-tTA/tetO-Cre and K-Ras$^{+/LSLG12Vgeo}$* mice were provided by M. Barbacid (CNIO, Madrid, Spain) (Guerra et al, 2007) and mated to generate *Elas-tTA/tetO-Cre;K-Ras$^{+/LSLG12Vgeo}$* (*Elast-K-Ras$^{G12V}$*). *Nlgn2$^{+/-}$* mice were provided by M. Missler (University of Munster, Germany). Pancreatitis was induced by intraperitoneal injection of 50 mg/ml caerulein (Sigma) once a day for the indicated duration. Experiments were performed according to the guidelines of the University of Torino (protocol approved: 864/2015).

### Probes and constructs

The five different *shRNA* MISSION RNA interference vectors for NLGN2 (TRCN0000075278, TRCN0000075279, TRCN0000075280, TRCN0000075281, and TRCN0000075282) and two for YAP (TRCN0000107266 and TRCN0000107267) were inserted into the pLKO.1-puro nonmammalian vector obtained from Sigma. Lentiviral particles were produced by transfecting 293T cells with the packaging system, target cells were transduced for 48 h with viral particles, and infected cells were selected with puromycin (2 μg/ml) for 5 days before seeding. The vectors carrying *shNLGN2* and *shYAP* featured puromycin resistance as a selection method; therefore we performed YAP silencing using an MOI = 1 (measured on puromycin-sensitive non-infected WT-HPDE) on mature confluent *shNLGN2* cells.

Stable silencing of targeted genes was analyzed by real-time PCR 7 and 30 days after infection, and the expression levels were compared to those measured in cells transduced with lentiviral vectors carrying the MISSION nontargeting *shRNA* control vector (Ctrl). NLGN2-MYC was synthesized by Invitrogen Gene Art service. The transgene was synthesized by inserting the MYC tag DNA sequence into the human *NLGN2* sequence immediately

downstream of position 1842, based on previous studies showing that inserting a tag in this position does not alter NLGN2 folding and function (Tsetsenis et al, 2014). The sequence was inserted into the pLenti6.3_V5-DEST vector and used for lentiviral production in 293T cells and stable cell infection. Infected cells were selected with blasticidin (2 µg/ml) for 5 days.

The following probes were used for the LACZ PCR assay: reverse LacZ (oIMR0040): CGTGGCCTGATTCATTCC forward LacZ (oIMR3054): ATCCTCTGCATGGTCAGGTC, LacZ amplicon: 315 bp.

The following probes were used for *NLGN2* exon3 splicing detection: probe a Forward exon 2: CCTGTACCTCAACCTCT ACGTG, probe "b" reverse exon3: TTGAGCGTCGCCTCGTC, probe "c" reverse exon 4 CATCGAACATGTTTCCGGTCC.

The following Taqman probes were chosen from Life Technologies: *hNLGN1* (Hs00995797_m1), *hNLGN2* (Hs00395803_m1), *hNLGN3* (Hs01043809_m1), *hNLGN4X* (Hs01934144_s1), *hNLGN4Y* (Hs01034378_s1), *hCCNB1* (Hs01030099_m1), *hCCNE1* (Hs01026536_m1), *hTBP* (Hs00427620_m1), *hAREG* (Hs00950669_m1), *hHBEGF* (Hs00181813_m1), *hCTGF* (Hs00170014_m1), *hCYR61* (Hs00155479_m1).

## Real-time PCR

Total RNA was extracted from cells with a Maxwell RNA extraction kit (Promega). RNA was quantified by a NanoDrop ND-100 spectrophotometer (NanoDrop Technologies). For cDNA synthesis, the High-Capacity cDNA Reverse Transcription Kit (Life Technologies) was used according to the manufacturer's instructions. Real-time PCR was performed on a CFX96 system (Bio-Rad) using TaqMan PCR Master Mix and probes (Thermo Fisher). All experiments were performed in triplicate.

## Antibodies

The following antibodies were used. The anti-pan-NLGN (L067, used for IP in murine cells) was obtained from New England Peptide. The antibodies against NLGN2 (No. 129 203; this product has been used for NLGN2 IP in cell lines, western blotting and human immunohistochemistry) and PALS1 (No. 220 003) were obtained from the Synaptic System. Neuroligin-2 (D-15) sc-14087 (this antibody was used for immunohistochemistry on frozen murine pancreas) and the anti-Vinculin (sc-7649) antibody was obtained from Santa Cruz. The antibodies against ZO1 (402200), PATJ (567033), and Ki67 (514520) were obtained from Life Technologies. The anti-GAPDH (ab8245), anti-Actin (ab179467), anti-cyclin E1 (ab3927), anti-AMOT (ab244534), and anti-DCAMKL1 (ab37994) antibodies were purchased from Abcam. The antibodies against YAP (#14074), YAP-phospho-S127 (#4911), E-cadherin (#3195), phospho-p44/42 MAPK (#4370), p44/42 MAPK (#9107), cyclin B1 (#4138), and the Myc tag (#2276) were purchased from Cell Signaling Technology.

## Coimmunoprecipitation

Mouse brains and cell cultures were homogenized in cold EB buffer (QIAGEN) and clarified by centrifugation. Protein extracts were incubated overnight with Dynabeads (Thermo Fisher) and preincubated with a primary antibody (1 µg per mg of protein).

## Western blot analysis

Cells were lysed in hot Laemmli buffer and sonicated. Cell lysis with cold EB buffer (QIAGEN) and clarification was optimal for NLGN2 western blot. About 30 µg Proteins were separated by electrophoresis on 4–20% SDS–polyacrylamide gels and transferred onto PVDF membranes. The membranes were incubated overnight at 4 °C with primary antibodies and were then incubated with horseradish peroxidase (HRP)-conjugated secondary antibodies (1:10,000; Jackson).

## Cell lines and cultures

WT-HPDE, 266-6 (ATCC), and K-RAS-HPDE cells (Siddiqui et al, 2018) were grown in RPMI 1640 medium, CACO2 cells (ATCC) were grown in DMEM, and human ECs (Nowak-Sliwinska et al, 2018) were grown in M199 medium. The culture media were supplemented with 10% fetal calf serum (FCS) and glutamine. For EK cell isolation from *Elast-K-Ras^{G12V}* mouse pancreata, tissues were manually minced, and cells were dissociated using a gentleMACS dissociator (Miltenyi Biotec) prior to incubation with 0.2 mg/ml COLLP-RO collagenase P from *Clostridium histolyticum* (Roche). Cells were frequently subcultured in DMEM, and different adhesion/detachment steps were used to remove fibroblasts. After one month, a pure culture of epithelial cells was obtained and characterized by *Krt19*, *Amy2a*, *Gfap*, *Ins*, and *Gcg* expression. When indicated, WT-HPDE cells were treated for 8 h with U0126 (10 µM) from Merck. To obtain confluent cultures, cells were seeded at 60% confluence and allowed to grow, and a one-half volume of fresh medium was added every two days. Experiments on mature confluent cells were performed after one week of visual identification of overgrowing *shNGN2* cells. All cell lines were routinely tested for mycoplasma contamination.

## Cell synchronization

Subconfluent (60–70% confluence) WT-HPDE cells were treated with Noco (50 ng/ml) or HU (500 mM) (Sigma) overnight to induce cell cycle arrest at the M or G1 phase, respectively. For HU treatment, cells were presynchronized by serum starvation for 24 h; then, HU was added to RPMI 1640 medium containing 10% FBS. For Noco treatment, mitotic cells detached by manual shaking were used. Thymidine was purchased from Sigma, and colcemid was purchased from Roche.

## Cystogenesis assay

For the cystogenesis assay, we adopted the "on-top" culture method (Lee et al, 2007). Briefly, six-well plates were coated with 50% growth factor-free Matrigel (Corning), which was allowed to solidify. Cells were detached and carefully checked for the formation of a single-cell suspension. Cells were suspended in 1 ml complete RPMI 1640 medium supplemented with 20 ng/ml EGF and seeded on top of the Matrigel layer. After cell sedimentation, an equal amount of RPMI 1640 medium supplemented with 20 ng/ml EGF plus 10% Matrigel (5% final) was added to the culture. The medium was changed every 2 days for the indicated duration. At the end of the experiment, Matrigel was dissolved in Cell Recovery Solution (Corning), and cysts were

collected by sedimentation, fixed with 4% paraformaldehyde, embedded in OCT compound, and frozen in cold isopentane. Cryosections were processed as detailed in the immunofluorescence protocol.

## Cytofluorimetry

Synchronized cells in 1 ml of 2 mM PBS-EDTA were fixed with 9 ml of cold 70% ethanol. Cells were left at −20 °C for 2 h and incubated with 0.1% Triton X-100 and 20 µg/ml propidium iodide (Life Technologies). Cells were analyzed using a BD FACS Canto II cytofluorimeter and FlowJo software.

## X-Gal, BrdU, and EdU reactivity

The following kits and reagents were used in our experiments. The X-Gal assay kit was obtained from Promega, the BrdU assay kit was obtained from Cell Signaling Technology, and the Click EdU imaging kit was obtained from Thermo Fisher. All kits were used according to the manufacturer's instructions. BrdU absorbance was measured with a GloMax Discover multiplate reader (Promega).

## Histology and immunoreactivity

Mouse pancreata were washed in PBS and were then fixed for 4 h in Zinc Fixative (BD Bioscience) and overnight in Zinc Fixative plus 30% glucose. Tissues were then embedded in OCT compound and snap frozen in isopentane cooled to −80 °C.

Cells were washed with $Ca^{2+}$ $Mg^{2+}$-supplemented PBS and fixed with 4% paraformaldehyde for 10 min. Cell samples and FFPE and OCT-embedded slices were permeabilized with PBS supplemented with 0.2% Triton X-100 (30 min at room temperature) and immersed in PBS containing 5% BSA and 5% donkey or rabbit serum according to the secondary antibody species. For immunohistochemistry, samples were treated with 3% $H_2O_2$ and then incubated overnight with the indicated primary antibodies at the dilution suggested by the manufacturers. After washing, the samples were incubated for 1 h at room temperature with a fluorescent secondary antibody (Alexa Fluor 555- or 488-conjugated, Life Technologies, [1:400]) and DAPI (Life Technologies, [1:1000]) for 1 h. For immunohistochemistry, samples were incubated with an HRP-conjugated rabbit anti-goat secondary antibody (DAKO, [1:400]) or anti-rabbit polymer (DAKO) for 1 h, and 3,3-diaminobenzidine tetrahydrochloride plus [1:1,000] was used as the chromogen (DAKO). Tissues were counterstained with hematoxylin.

## Survival analysis of PDAC patients

For PDAC gene expression analysis, we used a TCGA dataset (https://www.cancer.gov/tcga) containing 185 samples. Among 175 samples of tumors with the histological type "pancreatic adenocarcinoma", we used 168 samples in which *NLGN2* expression was annotated. Formalin-fixed, paraffin-embedded (FFPE) human samples were provided by the Applied Research Center (ARC-NET, University of Verona, Verona, Italy) and from the pathology files of the University of Torino at Città della Salute e della Scienza Hospital (Torino, Italy). Samples were collected with the approval of the Ethics Committee for Clinical Research of each institution with informed consent (protocols n. 26773 and n. 2CEI/ 543). Immunohistochemical observations of the NLGN2 signal in PDAC samples allowed us to discern strong positive and weak positive signals (the latter was visualized as a distinct but slightly increased signal compared with the background) as well as strong negative and weak negative signals (the latter was visualized as a signal comparable to or lower than the background). To analyze the clinical characteristics and survival of patients based on NLGN2 expression, the patients were categorized into two groups: those with negative NLGN2 expression and those with positive NLGN2 expression. The OS time was calculated by determining the time elapsed from the date of diagnosis until the date of death or last patient follow-up, and OS was analyzed by the Kaplan–Meier method and compared using the log-rank test.

The χ-square test was used to compare categorical variables among groups. To assess the prognostic impact of NLGN2 expression, univariate and multivariate analyses were conducted using the Cox regression model. The sample size was not predetermined, and the study was not designed to detect a specified effect size. All statistical tests were two-tailed, and differences with $p$ values less than 0.05 were considered statistically significant. Analyses were performed with IBM SPSS Statistics for Windows (Version 27.0).

## Transcriptome analysis and functional annotation

The total RNA from EK and 266-6 were subjected to high throughput sequencing for poly-A + RNAs by an external next-generation sequencing (NGS) facility (Fasteris, Geneva, Switzerland). RNA sequencing was performed on an Illumina NovaSeq 6000 platform, obtaining a mean of 20 million paired-end reads of 100 bps per sample. For both biological conditions, three replicates were analyzed. Initial quality check and processing was performed by Cutadapt (Martin, 2011), the processed reads were then mapped to reference genome (GRCm39) by STAR-2.5 aligner (Dobin et al, 2013), and quantification was performed by RSEM-1.3.0 (Li and Dewey, 2011) and ENSEMBL annotation. Functional annotation on the protein-coding genes of EK and 266-6 was performed by GSEA (Subramanian et al, 2005) desktop application from the BROAD Institute. The GSEA run against the canonical pathway geneset collection (m2.all.v2022.1) was performed with standard parameters by using 1000 geneset permutations. Only genesets with FDR ≤0.05 were considered. Part of the results shown in this study are based on MERAV (http://merav.wi.mit.edu).

## Data acquisition, quantification, and statistic

The WB immunoreactive bands were visualized by an enhanced chemiluminescence detection kit (Perkin-Elmer Life Science Products) and were quantified with Image Lab v5.2.1 software. Immunofluorescence images were acquired by using a Leica SPEII confocal laser-scanning microscope (Leica Microsystems). The same laser power, gain, and offset settings were maintained in multiple randomly chosen fields for image quantification with Fiji software. Statistical analysis were performed using GraphPad Prism 10 Software.

# Data availability

All raw sequencing data generated in this study have been submitted to the NCBI Gene Expression Omnibus (GEO; https://www.ncbi.nlm.nih.gov/sra/PRJNA1049617) and are available

under the accession numbers SRX22809790 (266-6 replicate 1), SRX22809791 (266-6 replicate 2), SRX22809792 (266-6 replicate 3), SRX22809793 (EK replicate 1), SRX22809794 (EK replicate 2), and SRX22809795 (EK replicate 3).

## Peer review information

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

## Acknowledgements

This work was supported by AIRC – Associazione Italiana per la Ricerca sul Cancro (grants 22910, 12182, and 18652), Ministero dell'Università e della Ricerca (PRIN 2017, grant 2017237P5X, PRIN 2022 grant 2022HREZJT, PNRR 3F 4 Health) FPRC 5xmille MIUR (Proactive) FB; AREM_RILO_19_01 - Bando Ricerca Locale 2019 to MA; AIRC (IG 19931), Fondazione Onlus Ricerca Molinette (Fondo CD38 and Fondo Ursula e Giorgio Cytron), Associazione Nastro Viola to FN; CRT (no. 2020.0719) to PC. The authors thank M. Barbacid and C. Guerra from the Spanish National Cancer Research Center (Madrid) for sharing the GEMM.

## Author contributions

**Emanuele Middonti**: Conceptualization; Investigation; Methodology; Writing—original draft; Writing—review and editing. **Elena Astanina**: Conceptualization; Investigation; Methodology. **Edoardo Vallariello**: Investigation; Methodology. **Roxana Maria Hoza**: Investigation; Methodology. **Jasna Metovic**: Resources. **Rosella Spadi**: Resources. **Carmen Cristiano**: Resources. **Mauro Papotti**: Resources; Writing—review and editing. **Paola Allavena**: Resources. **Francesco Novelli**: Conceptualization; Writing—review and editing. **Sushant Parab**: Formal analysis. **Paola Cappello**: Conceptualization; Writing—review and editing. **Aldo Scarpa**: Resources. **Rita Lawlor**: Resources. **Massimo Di Maio**: Formal analysis; Writing—review and editing. **Marco Arese**: Conceptualization; Methodology; Writing—review and editing. **Federico Bussolino**: Conceptualization; Supervision; Funding acquisition; Methodology; Writing—review and editing.

## Disclosure and competing interests statement

The authors declare no competing interests.

# Expanded View Figures

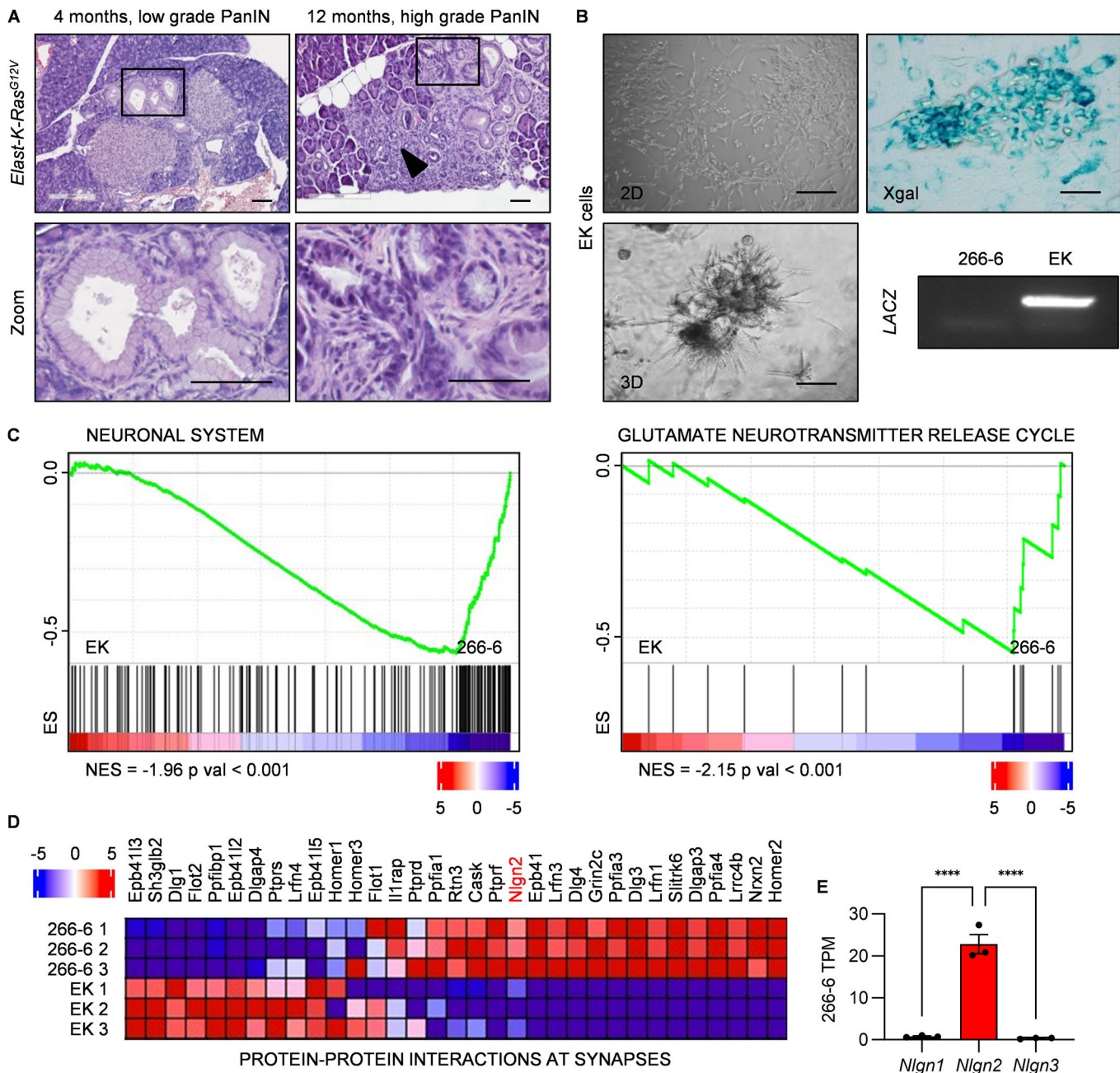

Figure EV1. Characterization of PanIN lesions in *Elast-K-Ras^G12V* mice and characterization of EK cells.

(A) *Elast-K-Ras^G12V* mice spontaneously develop PanIN lesions with progressive features. The upper panel indicates PanIN grade progression with mouse age. At month 12, histological signs of pancreatitis were present (arrowhead). The square boxes indicate the areas shown at higher magnification in the bottom panels. In low-grade PanIN, mucin accumulation was still evident, and in high-grade PanIN, progressive loss of polarity was observed. (B) Left panels: representative images of EK cells growing with a mesenchymal phenotype on a 2D plastic surface (top) and with disordered and unpolarized architecture in 3D Matrigel culture (bottom). Right panels: EK cells carry *K-Ras^G12V*, as demonstrated by the expression of LACZ demonstrated by X-Gal staining (top) and by PCR (bottom). (C) "Neuronal system" and "Glutamate neurotransmitter release cycle" gene sets enriched in 266-6 compared to EK cells. Red and blue color scale indicate $\log_2$ mRNA fold change ES: enrichment Score. NES: normalized enrichment score. (D) Heatmap showing the $\log_2$ mRNA fold change (red and blue scale) between EK and 266-6 cells of genes in "protein–protein interactions at synapses" gene set enriched in 266-6 according to GSEA. *Nlgn2* is highlighted in red. (E) TPM count of *Nlgn1*, *Nlgn2*, and *Nlgn3* transcripts from RNA seq data of 266-6 cells, showing that *Nlgn2* is the most expressed. Data information: (A) representative images of 12 mice, scale bar: 0.1 mm. (B) Scale bars: 0.05 mm. (C) Gene sets are identified by GSEA (Kolmogorov–Smirnov test). (E) The graph shows mean TPM count ± SEM (*n* of biological replicates = 3; unpaired *t*-test: ****$p < 0.0001$).

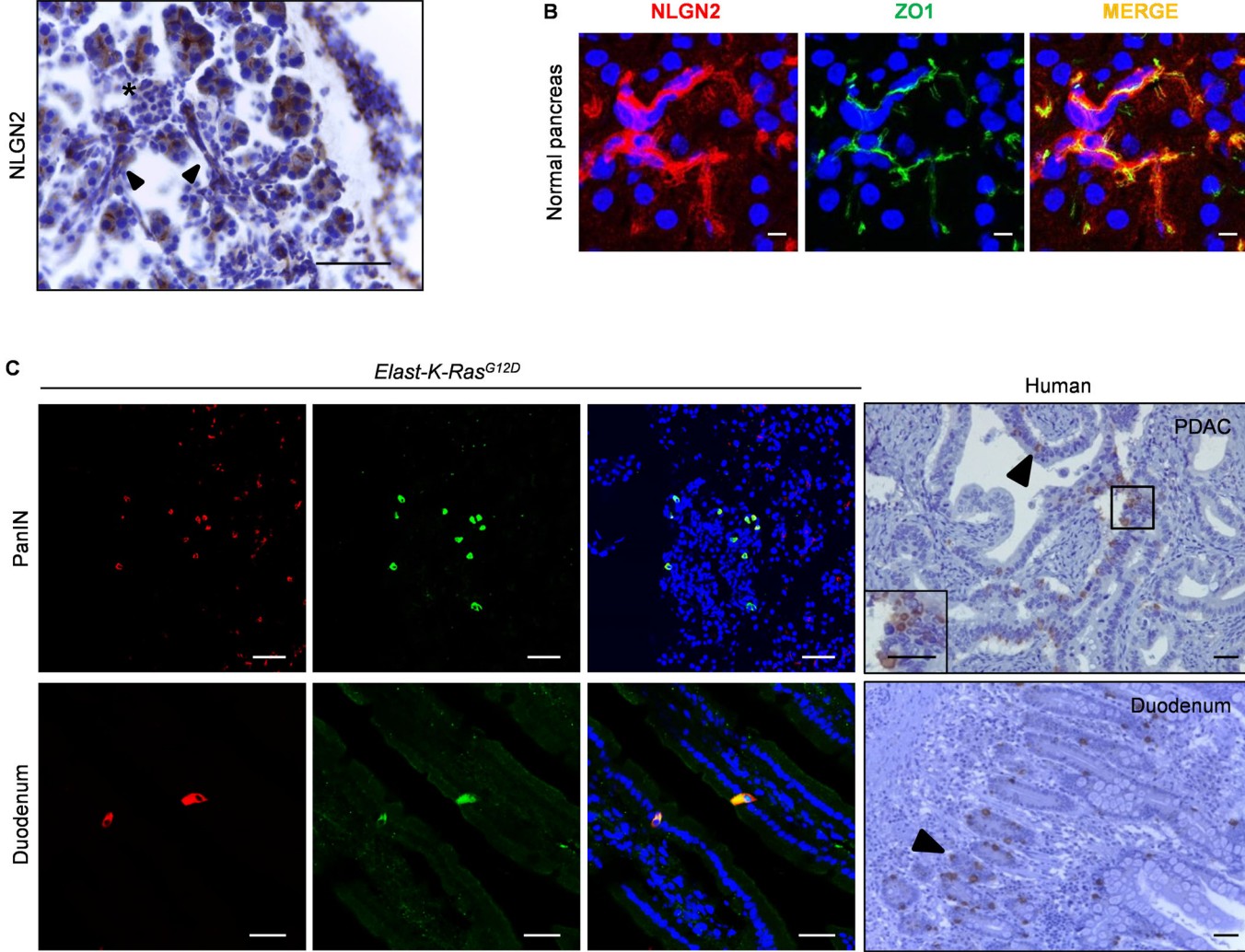

**Figure EV2. NLGN2 expression pattern.**

(A) Immunohistochemical analysis of a pancreas from a 6-month-old *Elast-K-Ras*⁺/⁺ mouse showing NLGN2 localization in the pancreatic duct (full arrowhead) and pancreatic insulae (asterisk). (B) Immunofluorescence analysis of NLGN2 localization in the pancreas of 6-month-old mice. The images are representative of ductal cells of *Elast-K-Ras*⁺/⁺ mice and untransformed ductal cells of *Elast-K-Ras*^G12V mice, which have superimposable aspects. NLGN2 and ZO1 are colocalized in the apical portion of pancreatic ductal cells. (C) Presence of a subpopulation expressing high amounts of NLGN2 in *Elast-K-Ras*^G12V mice (left, immunofluorescence) and in human PDAC specimens (right, immunohistochemistry). Left panels: immunofluorescence staining of PanIN (top) and duodenum tuft cells (bottom) in *Elast-K-Ras*^G12V mice showing a cell subpopulation expressing high levels of NLGN2, which costained with DCLK1. Right panels: immunohistochemistry revealed an NLGN2-overexpressing cell subpopulation (arrowheads) in human PDAC tissues (top, inset shows higher magnification of NLGN2⁺ cells) and in the normal duodenum (bottom). Data information: (A) scale bar: 0.1 mm; (B) scale bar: 10 μm; (C) scale bars: 50 μm in mouse specimens; 0.1 mm in human specimens.

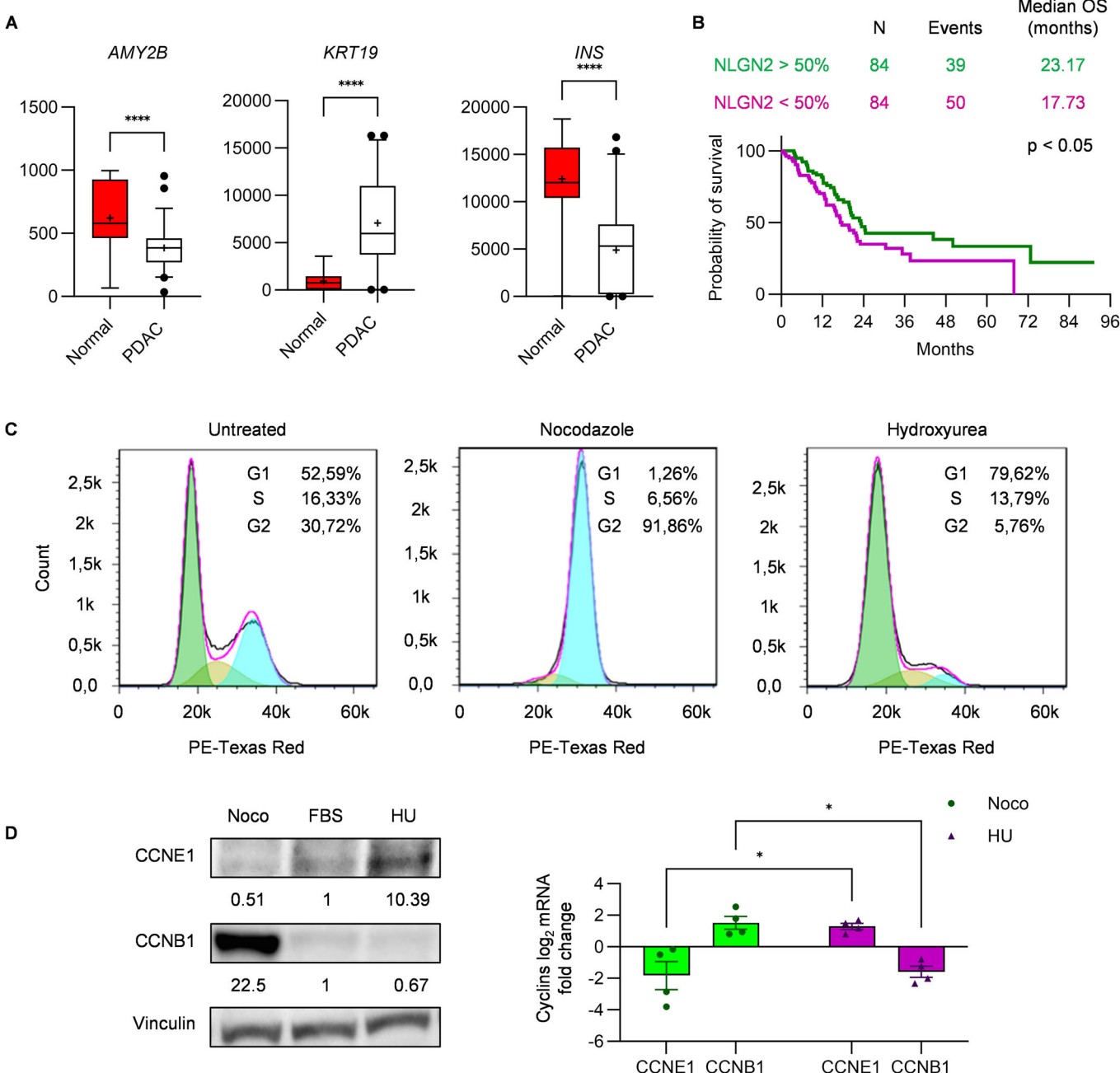

**Figure EV3. Analysis of NLGN2 expression in public datasets.**

(A) Relative differential expression of *AMY2B*, *KRT19*, and *INS* between normal pancreas and PDAC tissues as determined in the MERAV dataset. (B) Kaplan–Meier survival curves for *NLGN2* mRNA expression in the TCGA cohort. The panel indicates the number of events (deaths) and the median survival time, the graph show the survival probability of patients divided in half by *NLGN2* expression. (C) WT-HPDE cell synchronization was validated through FACS analysis, showing the DNA content (*n* = green peak; 2*n* = blue peak) in untreated, Noco-treated, and HU-treated cells. The percentage of cells synchronized in the G1, S, or G2 phase is embedded in each graph. (D) Validation of cells synchronized in G1, S, or G2 phase through measurement of cyclin E1 and B1 expression by Western blotting, reporting relative densitometry normalized on Vinculin (left) and real-time PCR (right). Data information: in (A) the center lines show the medians and the crosses indicate sample means; box limits indicate the 25th and 75th percentiles; the whiskers extend from the 5th and 95th percentile values and outliers are represented by dots; $n = 57$ PDAC; 16 Normal pancreas. (unpaired *t*-test: ****$p < 0.0001$). (B) OS was analyzed by the Kaplan–Meier method and compared using the log-rank test ($p < 0.05$). In (D) the graph shows mean mRNA $\log_2$ fold change ± SEM compared to untreated cells. Each gene was analyzed in triplicate, four biological replicates (multiple unpaired *t*-test: *$p < 0.05$). Source data are available online for this figure.

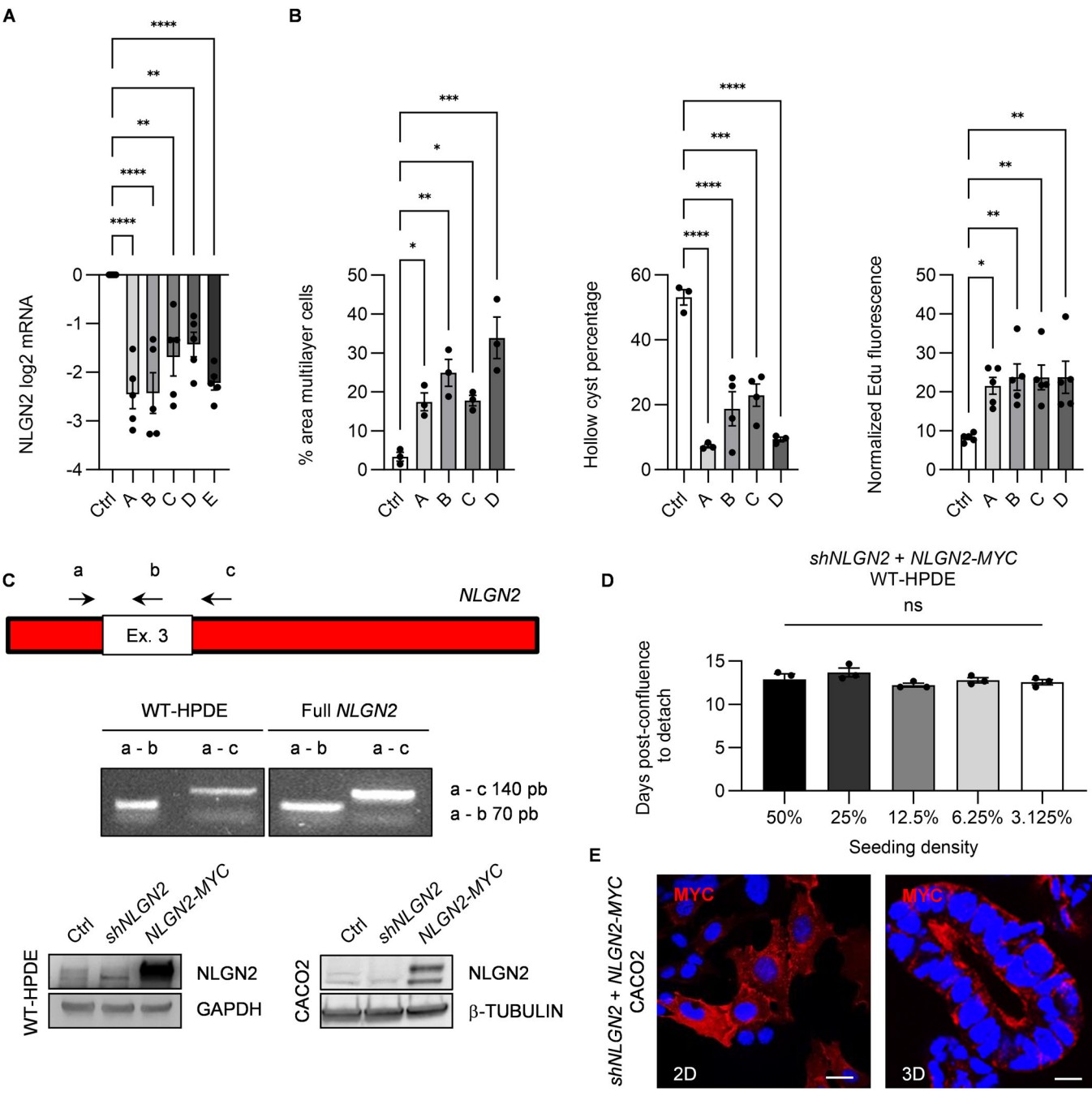

**Figure EV4. Rescue expression of NLGN2-MYC in *shNLGN2* cells.**

(A) Silencing efficacy of five different short hairpin RNA for *NLGN2* (A–E) as shown by reduced *NLGN2* mRNA expression. (B) Reproducibility validation of four different short hairpin RNA for *NLGN2* (A–D) on cell phenotype compared to Ctrl WT-HPDE. Left: the area percentage of overgrowing cells in mature confluent *shNLGN2* WT-HPDE was quantified. Mid: Cyst morphology quantification indicating the percentage of hollow cysts. Right: the graph shows EdU fluorescence normalized to the number of nuclei. (C) Top: Schematic representation of the PCR strategy used to identify the *NLGN2* isoform in WT-HPDE cells. The white inset represents the alternatively spliced third exon in the *NLGN2* sequence (Ex. 3). Primer pairing sites and directions are depicted with directional arrows and designated "a", "b", and "c". Mid: PCR products were obtained using primer pairs "a" and "b" or "a" and "c" on WT-HPDE cell cDNA or the recombinant full *NLGN2* control sequence, respectively. Bottom: Western blot in WT-HPDE and CACO2, showing reduced NLGN2 expression due to transduction of the *shNLGN2* carrying vector and NLGN2 re-expression in *shNLGN2* cells after transduction with the NLGN2-MYC lentiviral vector. (D) *shNLGN2 WT-HPDE* + NLGN2-MYC cells were seeded at different density in linear dilution, to exclude the consequences of the time needed to reach confluence. The graph shows the number of days required for detachment after confluence. (E) Expression of MYC-tagged NLGN2 in Ctrl and *shNLGN2* CACO2 cells is shown. The membrane and apical localization of exogenous NLGN2 was revealed using an anti-MYC antibody in 2D and 3D CACO2 cells. Data information: in (A) the graph reports $\log_2$ mRNA fold change ± SEM compared to Ctrl WT-HPDE of triplicate samples in five experiments (one-way ANOVA: **$p < 0.01$; ****$p < 0.0001$). In (B) left: mean ± SEM of 3 biological replicates (one-way ANOVA: *$p < 0.05$; **$p < 0.01$; ***$p < 0.001$). Mid: mean ± SEM of 3 or 5 biological replicates (one-way ANOVA: ***$p < 0.001$; ****$p < 0.0001$). Right: mean ± SEM of 5 biological replicates (one-way ANOVA: *$p < 0.05$; **$p < 0.01$). In (D) the graph reports mean ± SEM, biological replicates = 3 (one-way ANOVA: non significant: ns). (E) Scale bar: 20 μm. Source data are available online for this figure.

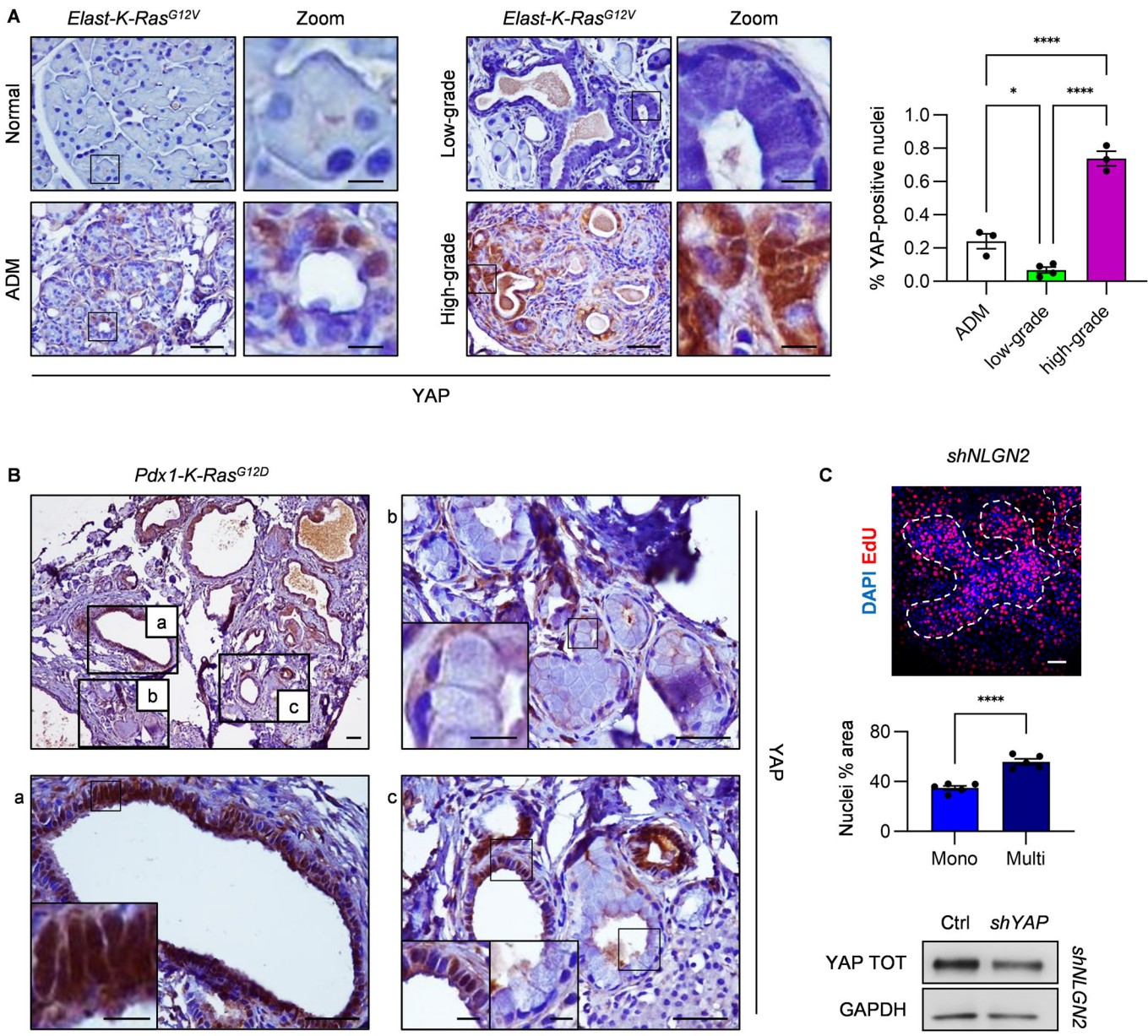

**Figure EV5. YAP expression in murine ADM and PDAC.**

(**A**) Immunohistochemical analysis of total YAP in normal pancreas, ADM, low-grade PanIN, and high-grade PanIN in 12-month-old *Elast-K-Ras*^G12V^ mice. The right panels show higher-magnification versions of the corresponding images (Zoom). The graph shows the percentage of YAP-positive nuclei. (**B**) Immunohistochemical analysis of total YAP in PDAC tissues of *Pdx1-K-Ras* mice. Top-left panel: the gross appearance of PDAC tissues. a: high-grade PanIN; b: low-grade PanIN; c: glands with mixed features of polarized and crowded cells exhibiting different degrees of YAP nuclear localization. (**C**) Top: Immunofluorescence of nuclei (DAPI) and EdU in *shNLGN2* mature confluent WT-HPDE. White lines distinguish monolayer and multilayer areas and highlight heterogeneity in *shNLGN2* culture. Mid graph: DAPI area in monolayer and multilayer area; Bottom: total YAP reduction in *shNLGN2 + shYAP* mature confluent WT-HPDE. Data information: in (**A**) scale bar: 50 µm, Zoom scale bar: 10 µm. The graph shows mean percentage ± SEM; *n* mice with ADM and high-grade PanIN = 3, *n* mice with low-grade PanIN = 4; *n* of YAP-positive nuclei (ADM: *n* = 424; low-grade PanIN: *n* = 783; high-grade PanIN: *n* = 286); One-way ANOVA: *$*p < 0.05$; ****$p < 0.0001$. (**B**) Scale bar: 0.2 mm, (a–c) scale bar: 50 µm, inset scale bar: 10 µm. (**C**) Scale bar: 50 µm; the graph reports mean ± SEM, biological replicates = 5, unpaired *t*-test: ****$p < 0.0001$. Source data are available online for this figure.

