## [Peer Review File · EMBO Reports]

A Neuroligin-2-YAP axis regulates progression of pancreatic intraepithelial neoplasia

Emanuele Middonti, Elena Astanina, Edoardo Vallariello, Roxana Maria Hoza, Jasna Metovic, Rosella Spadi, Carmen Cristiano, Mauro Papotti, Paola Allavena, Francesco Novelli, Sushant Parab, Paola Cappello, Aldo Scarpa, Rita Lawlor, Massimo Di Maio, Marco Arese, and Federico Bussolino

Corresponding author(s): Federico Bussolino (federico.bussolino@unito.it) , Emanuele Middonti (emanuele.middonti@unito.it)

Review Timeline:

Submission Date:	5th Jun 23
Editorial Decision:	12th Jul 23
Revision Received:	21st Nov 23
Editorial Decision:	24th Jan 24
Revision Received:	5th Feb 24
Accepted:	13th Feb 24

Editor: Achim Breiling

Transaction Report:

Dear Prof. Bussolino,

Thank you for the submission of your manuscript to EMBO reports. I have now received the reports from the three referees that were asked to evaluate your study, which can be found at the end of this message.

As you will see, the referees think that these findings are of high interest. However, they have several comments, concerns, and suggestions (mainly referees #2 and #3), indicating that a major revision of the manuscript is necessary to allow publication of the study in EMBO reports. As the reports are below, and all the referee concerns need to be addressed, I will not detail them here.

Given the constructive referee comments, I would like to invite you to revise your manuscript with the understanding that all referee concerns must be addressed in the revised manuscript and in a detailed point-by-point response. Acceptance of your manuscript will depend on a positive outcome of a second round of review. It is EMBO reports policy to allow a single round of revision only and acceptance of the manuscript will therefore depend on the completeness of your responses included in the next, final version of the manuscript.

- 1) a .docx formatted version of the final manuscript text (including legends for main figures, EV figures and tables), but without the figures included. Figure legends should be compiled at the end of the manuscript text.
- 2) individual production quality figure files as .eps, .tif, .jpg (one file per figure), of main figures (up to 8) and EV figures (up to 5). Please upload these as separate, individual files upon re-submission.

- 4) a complete author checklist, which you can download from our author guidelines

(<https://www.embopress.org/page/journal/14693178/authorguide>). Please insert page numbers in the checklist to indicate where the requested information can be found in the manuscript. The completed author checklist will also be part of the RPF.

5) that primary datasets produced in this study (e.g. RNA-seq, ChIP-seq, structural and array data) are deposited in an appropriate public database. If no primary datasets have been deposited, please also state this in a dedicated section (e.g. 'No primary datasets have been generated and deposited'), see below.

The accession numbers and database should be listed in a formal "Data Availability" section (placed after Materials & Methods) that follows the model below. This is now mandatory (like the COI statement). Please note that the Data Availability Section is restricted to new primary data that are part of this study. This section is mandatory. As indicated above, if no primary datasets have been deposited, please state this in this section

Data availability

8) Regarding data quantification and statistics, please make sure that the number "n" for how many independent experiments were performed, their nature (biological versus technical replicates), the bars and error bars (e.g. SEM, SD) and the test used to calculate p-values is indicated in the respective figure legends (also for potential EV figures and all those in the final Appendix). Please also check that all the p-values are explained in the legend, and that these fit to those shown in the figure. Please provide statistical testing where applicable. Please avoid the phrase 'independent experiment', but clearly state if these were biological or technical replicates. Please also indicate (e.g. with n.s.) if testing was performed, but the differences are not significant. In case n=2, please show the data as separate datapoints without error bars and statistics. See also: <http://www.embopress.org/page/journal/14693178/authorguide#statisticalanalysis>

9) Please add scale bars of similar style and thickness to microscopic images, using clearly visible black or white bars (depending on the background). Please place these in the lower right corner of the images themselves. Please do not write on or near the bars in the image but define the size in the respective figure legend.

10) Please also note our reference format:

12) We now use CRediT to specify the contributions of each author in the journal submission system. CRediT replaces the author contribution section. Please use the free text box to provide more detailed descriptions and do not provide your final manuscript text file with an author contributions section. See also our guide to authors: <https://www.embopress.org/page/journal/14693178/authorguide#authorshipguidelines>

Please order the manuscript sections like this, using these names:

Title page - Abstract - Keywords - Introduction - Results - Discussion - Materials and Methods - Data availability section - Acknowledgements - Disclosure and Competing Interests Statement - References - Figure legends - Expanded View Figure legends

Finally, please note that all corresponding authors are required to supply an ORCID ID for their name upon submission of a revised manuscript. Please find instructions on how to link the ORCID ID to the account in our manuscript tracking system in our Author guidelines: <http://www.embopress.org/page/journal/14693178/authorguide#authorshippinguidelines>

I look forward to seeing a revised version of your manuscript when it is ready. Please let me know if you have questions or comments regarding the revision.

Yours sincerely,

Referee #1:

This is an extremely interesting MS covering for the first time a new gene, NLGN2, that is also a new marker and a new functional player, in pancreatic biology and tumorigenesis. Expressed in the normal pancreas, NLGN2 consistently decreases in low grade PanINs and then PDACs. Data from real tumors are convincing. The MS offers a very solid set of redundant NLGN2 inactivation approaches, including a tour-de-force through several mouse models and in vitro manipulations, compellingly showing a role for NLGN2 in pancreatic exocrine cell biology and that NLGN2 loss causes aberrant epithelial cystogenesis and proliferation. Mechanistically they link, quite convincingly, NLGN2-related processes to YAP regulation showing that NLGN2 drives cell polarization and control the PALS1/PATJ complex, that are lost in cancer, including PanIN and PDACS and that are known to be, so far not very clearly, linked to YAP stabilization. All in all, I found the MS very refreshing and well organized; individual claims are supported from different, redundant and converging lines of evidence.

I frankly do not see major obstacles to a prompt publication of this work. If revision is requested by other reviewers, I only advise the authors to cite earlier literature, in particular the pioneering work of Wrana and Varelas on Crumbs, and of Cordenonsi et al on YAP and Scribble polarity complexes in mammary tumor cell plasticity and stemness.

Referee #2:

In the manuscript by Middonti et al, the authors investigate the relevance of neuronal gene expression dysregulation in pancreatic cancer (PDAC). The authors present evidence that NLGN2, a known regulator of synapse biology and cell polarity, is expressed in the exocrine pancreas and regulates contact inhibition and epithelial polarity. They find that NLGN2 becomes silenced during PDAC progression to promote loss of cell polarity. Mechanistically, they link the function of NLGN2 to a role in regulating PALS1/PATJ complex, which in turn regulates YAP.

A strength of this study would be its conceptual novelty, linking neuronal gene expression with epithelial cancer biology. Another strength would be the clinical correlations with the observations in experimental systems. A major weakness of this work is technical: the use of shRNA alone to study NLGN2-dependent phenotypes without appropriate controls. In addition, the evidence presented is quite limited regarding the proposed mechanism (PALS1/PATJ and YAP as downstream mediators of NLGN2 function).

Major points:

1) The phenotypic data presented in Figure 4 is interesting, but I have a concern about off-target effects of these shRNA used here accounting for the observed phenotypes. I ask the authors here to a) validate their findings in Figure 4 using CRISPR-based knockout NLGN2 with at least two independent sgRNAs and b) rescue these effects with a CRISPR-resistant cDNA.

2) The mechanism proposed does not include sufficient evidence of causality (figure 5). NLGN2 KD is shown to alter YAP and PALS1/PATJ activity/expression, but there is no evidence provided that these two regulators are relevant to the phenotypic effects of silencing NLGN2. To address this and support the proposed mechanism, the authors should perform rescue experiments that seek to manipulate YAP or PALS1/PATJ and determine whether such manipulations can rescue the phenotypes observed upon knocking down NLGN2 (assay shown in Figure 4). I consider such epistasis experiments to be critical evidence needed to support the mechanism the authors are proposing as the main claim of this study.

Referee #3:

In this paper, Emanuele Middonti and colleagues present data trying to elucidate the mechanisms underlying the progression from low- to high-grade PanINs. They put forth the intriguing hypothesis that the synaptic molecule NLGN2 is expressed in pancreatic exocrine cells and may play a pivotal role in regulating contact inhibition and epithelial polarity. Moreover, they suggest that NLGN2 localizes at tight junctions in acinar cells, exhibits diffuse cytosolic distribution in low-grade PanINs, and becomes lost in high-grade PanINs. Importantly, they propose a mechanistic link wherein NLGN2 should be crucial for the formation of the PALS1/PATJ complex, which subsequently should induce contact inhibition by modulating YAP function.

While this study presents intriguing insights, it could benefit with a greater coherence, rigor, and most notably, the inclusion of a working hypothesis.

Comments:

What is the rationale behind using a panel of 80 genes selected for their involvement in synaptogenesis, axon guidance, neuronal growth, and development? Why have the authors limited themselves to this group of genes? Is there a hypothesis driving this selection? If not, restricting the analysis to these genes could be considered as a flawed strategy when aiming to explain the mechanism underlying the transition from low- to high-grade PanINs.

The use of EK and 266.6 cells, which are from completely different origins, as cellular models is not acceptable for screening purposes.

Loss of NLGN2 expression appears to be associated with EMT transition. It would be worthwhile to study the correlation between the expression of EMT markers (E-cadherin, N-cadherin, Vimentin, SNAIL, SLUG, and ZEB1) and NLGN2.

Is there a correlation between expression of GATA6 and NLGN2?

PDAC can be classified into classical or basal-like subtypes, which perfectly explain patient survival outcomes. The association observed by the authors in the TCGA cohort may simply reflect the loss of NLGN2 expression as the pancreatic epithelial cell undergoes dedifferentiation. Please apply the Purist classifier and examine how each subtype expresses NLGN2.

Some of the results in this study are repetitions or confirmations of previously published works, particularly those involving CoIP.

The Results section should focus solely on objective description of data. In the current version, the results are systematically introduced and discussed, making the reading lengthy and tedious. The presence of references in the Results section should be exceptional and justified.

To my known the aim of this study was to explain the mechanisms involved in the progression from low- to high-grade PanINs. It seems unlikely that any clues or answers to this question have been found in the end.

Point-by-point response.

Referee #1:

This is an extremely interesting MS covering for the first time a new gene, NLGN2, that is also a new marker and a new functional player, in pancreatic biology and tumorigenesis. Expressed in the normal pancreas, NLGN2 consistently decreases in low grade PanINs and then PDACs. Data from real tumors are convincing. The MS offers a very solid set of redundant NLGN2 inactivation approaches, including a tour-de-force through several mouse models and in vitro manipulations, compellingly showing a role for NLGN2 in pancreatic exocrine cell biology and that NLGN2 loss causes aberrant epithelial cystogenesis and proliferation. Mechanistically they link, quite convincingly, NLGN2-related processes to YAP regulation showing that NLGN2 drives cell polarization and control the PALS1/PATJ complex, that are lost in cancer, including PanIN and PDACS and that are known to be, so far not very clearly, linked to YAP stabilization. All in all, I found the MS very refreshing and well organized; individual claims are supported from different, redundant and converging lines of evidence.

I frankly do not see major obstacles to a prompt publication of this work. If revision is requested by other reviewers, I only advice the authors to cite earlier literature, in particular the pioneering work of Wrana and Varelas on Crumbs, and of Cordenonsi et al on YAP and Scribble polarity complexes in mammary tumor cell plasticity and stemness.

We thanks this referee for the shared enthusiasm on these findings. We added the suggested citations (page 4, line 80; and page 4, line 83)

Referee #2:

In the manuscript by Middonti et al, the authors investigate the relevance of neuronal gene expression dysregulation in pancreatic cancer (PDAC). The authors present evidence that NLGN2, a known regulator of synapse biology and cell polarity, is expressed in the exocrine pancreas and regulates contact inhibition and epithelial polarity. They find that NLGN2 becomes silenced during PDAC progression to promote loss of cell polarity. Mechanistically, they link the function of NLGN2 to a role in regulating PALS1/PATJ complex, which in turn regulates YAP.

A strength of this study would be its conceptual novelty, linking neuronal gene expression with epithelial cancer biology. Another strength would be the clinical correlations with the observations in experimental systems. A major weakness of this work is technical: the use of shRNA alone to study NLGN2-dependent phenotypes without appropriate controls. In

addition, the evidence presented is quite limited regarding the proposed mechanism (PALS1/PATJ and YAP as downstream mediators of NLGN2 function).

We thank this referee for his/her positive comments and according to the suggestions proposed we tried to improve the meaning of our results by performing in vitro experiments planned to demonstrate that YAP is an effector of NLGN2.

Major points:

1) The phenotypic data presented in Figure 4 is interesting, but I have a concern about off-target effects of these shRNA used here accounting for the observed phenotypes. I ask the authors here to a) validate their findings in Figure 4 using CRISPR-based knockout NLGN2 with at least two independent sgRNAs and b) rescue these effects with a CRISPR-resistant cDNA.

We agree that the possible off-target of shRNA is reason of concerns. Due to the time required to obtain NLGN2 guides, to generate cell clones, validate the effective NLGN2 KO, and perform a sufficient number of experiments suitable for statistical analysis, we exploited another approach to validate the specificity of our results. As already indicated in the Supplemental material and method (Probes and constructs section, now in Appendix) in the old submission we used 5 different shRNA targeting NLGN2 in different regions. Our preliminary analysis had already resulted in similar phenotypes among the 5 shRNAs, before the initial submission. To further address this point, we repeated the experiments to demonstrate a statistical significance between cells carrying empty vector and cells infected with vectors encoding the 5 shRNAs (Fig. EV4A, B). To further support that the observed phenotypes depended on NLGN2 silencing, we demonstrated that HPDE express only NLGN2 among NLGNs, excluding the implication of other NLGNs (Table 3). We also reported that exogenous overexpression of NLGN2-MYC in shNLGN2 WT-HPDE triggered cell detachment from the culture dish before they reached mature confluence, while these cells do not show any gross alteration at subconfluence and early confluence (Fig EV4D). This demonstrates NLGN2-MYC specifically rescued the loss of sensitivity to crowding in shNLGN2 cells, which is instrumental for polarization and contact inhibition. Moreover, NLGN2-MYC expression in shNLGN2 CACO2 cells significantly rescued cyst morphology and polarization (Fig 4E, F), indicating that these effects are at least partially regulated by NLGN2. Even in absence of experiments performed in cells in which NLGN2 was deleted by CRISPR/CAS genome editing we strongly believe that the results shown are effectively due to the NLGN2 down-modulation.

2) The mechanism proposed does not include sufficient evidence of causality (figure 5). NLGN2 KD is shown to alter YAP and PALS1/PATJ activity/expression, but there is no evidence provided that these two regulators are relevant to the phenotypic effects of silencing NLGN2. To address this and support the proposed mechanism, the authors should perform rescue experiments that seek to manipulate YAP or PALS1/PATJ and determine whether such manipulations can rescue the phenotypes observed upon knocking down NLGN2 (assay shown in Figure 4). I consider such epistasis experiments

to be critical evidence needed to support the mechanism the authors are proposing as the main claim of this study.

We agree with referee's statement that correlation does not necessarily mean causation. We excluded to rescue shNLGN2 phenotype manipulating PATJ/PALS1, as this would mean to overexpress them, while it is thoroughly reported that overexpression of polarity cues leads to their mislocalization and altered polarity (reviewed by Halaoui, R. & McCaffrey, L. Rewiring cell polarity signaling in cancer. Oncogene 34, 939-950 (2015)). Therefore, we planned to rescue the phenotype observed in mature confluent shNLGN2 cells by silencing YAP. In shNLGN2 cells we reported an heterogeneous phenotype consisting of at least two distinct cell subpopulations: a first population of cells growing in monolayer with expression of PATJ and low level of proliferation; a second population growing in multilayer, lacking PATJ and with high level of proliferation (Fig. 5I, Fig. EV5C). YAP silencing induced in shNLGN2 cells at mature confluence seems to specifically affect multilayered cells, inducing their detachment with appearance of "floating" cell rafts that are absent in shNLGN2 cells alone. We ensured that these floating cell rafts were actually dying and were not able to grow further colonies. Of note, Ctrl and monolayer shNLGN2 cells were unaffected by YAP silencing (Fig. 5J).

To quantify the rate of cell detachment we tried to obtain a single cell suspension to be analyzed by cytofluorimetry without success. The cell rafts observed in YAP and NLGN2 silenced cells tightly stuck together, and enzymatic or mechanic attempts to obtain a single cell suspension resulted in cell breaking. Therefore, we overcame this problem by a manual count of trypan blue positive cells. Quantitatively we observed an increase in cell death from 5.1% of total cells (mono and multilayer) in shNLGN2 to 20.9% in shYAP+shNLGN2 cells. Overall, YAP silencing is able to rescue the phenotype observed in mature confluent shNLGN2 cells (page 11, lines 330-341)

We already have preliminary data from single cell sequencing regarding the emergence of an heterogeneous expression pattern in shNLGN2 mature confluent cells. This subgroup corresponds to the over-growing cells forming a multilayer cells and show transcript expression suggestive to YAP activation in multilayer cells.

Referee #3:

In this paper, Emanuele Middonti and colleagues present data trying to elucidate the mechanisms underlying the progression from low- to high-grade PanINs. They put forth the intriguing hypothesis that the synaptic molecule NLGN2 is expressed in pancreatic exocrine cells and may play a pivotal role in regulating contact inhibition and epithelial polarity. Moreover, they suggest that NLGN2 localizes at tight junctions in acinar cells, exhibits diffuse cytosolic distribution in low-grade PanINs, and becomes lost in high-grade PanINs. Importantly, they propose a mechanistic link wherein NLGN2 should be crucial for the formation of the PALS1/PATJ complex, which subsequently should induces contact inhibition by modulating YAP function.

While this study presents intriguing insights, it could benefit with a greater coherence, rigor, and most notably, the inclusion of a working hypothesis.

Comments:

What is the rationale behind using a panel of 80 genes selected for their involvement in synaptogenesis, axon guidance, neuronal growth, and development? Why have the authors limited themselves to this group of genes? Is there a hypothesis driving this selection? If not, restricting the analysis to these genes could be considered as a flawed strategy when aiming to explain the mechanism underlying the transition from low- to high-grade PanINs.

We agree that a screening analysis on a selected number of genes is biased and may hide further discoveries in this field. Therefore, we performed RNA-seq on EK and 266-6 cells. First, we confirmed that the sequencing results are comparable with our panel analysis. We show here for clarity the comparison between the significant differential expressed genes (DEGs) shown in the Table 1 of the old version with those obtained from RNA seq analysis

Gene Symbol	log ₂ FoldChange Panel	log ₂ FoldChange Seq	Adjusted p.value Seq
Ntrk1	-9.965784285	-10.10812396	8.64E-17
Ptfla	-9.965784285	-13.22777527	2.46E-28
Gfra3	-9.965784285	-7.892109381	7.84E-10
Nr2e1	-8.380821784	-5.990164523	6.24E-05
Nlgn1	-6.795859283	-7.98412667	6.00E-10
Sema3g	-5.795859283	-4.499638229	4.37E-31
Sema4a	-5.573466862	-6.819451807	8.79E-149
Plxnc1	-4.506352666	-7.723778626	6.54E-13
Gfra1	-4.035046947	-5.388938949	0
Plxna3	-3.426625474	-3.863035673	0
Gphn	-2.164884385	-1.419754807	1.35E-68
Pspn	-1.932361283	-0.58622871	0.811497
Nlgn3	-1.314732593	-2.843581236	0.00227
Nlgn2	-0.751465164	-0.725708851	3.03E-24
Foxa2	-0.701341684	-0.281242174	8.48E-05
Plxnb2	0.818032475	1.194808584	1.36E-149
Nes	1.794103899	4.041285217	0
Sema3c	2.271425676	4.220694431	0
Rest	2.638305577	1.999194075	7.00E-157
Bdnf	5.072363095	8.823595042	1.63E-79
Plxnb3	5.885354722	6.201177419	2.58E-64

Gdnf	7.920739904	11.83025264	9.57E-23
Sema3e	12.93808616	11.42724447	5.75E-82

GSEA analysis revealed set of neural genes enriched in 266-6 compared to EK, including “protein-protein interactions at synapses”. As introduced in the main text (page 3: lines 69-70, and page 5: lines 118-120), due to the parallelism between synapses and tight junction, and being synapses structure highly specialized in polarization, we think that the loss of this gene enriched set is fascinating and further justifies our study on NLGN2.

According to this transcriptomic analysis we performed the following changes in the text. First, in Fig. 1A we added GSEA analysis showing “protein-protein interactions at synapses”, in which *Nlgn2* was present, and we added differential TPM counts of *Nlgn2* between 266.6 and EK cells. In figure EV1 C and D we added Neural related gene set found by GSEA analysis and heat map of “protein-protein interactions at synapses”. Second, we removed panel A of old Fig 1 showing that *Nlgn2* is the most expressed *Nlgn* in 266-6, this has been substituted by TPM counts in Fig. EV1E. For space limitation we also removed from Fig. 1A the panel showing the brain lysates used for quality check of NLGN2 immunoprecipitation. Actually this control did not add any relevant information, but we added raw data of this experiment as requested. Third, because the reader can access the total DEG from RNA seq, we removed Fig. S1C, which showed the reduction of *Amy2a* and increased *CK19* in EK. Actually, DEGs between 266-6 and EK transcriptomes clearly demonstrated the loss of acinar markers in EK cells and the acquisition of ductal markers (page 5: lines 112, 114). The transcriptome analysis of EK cells revealed the lack of *Ins*, *Gcg*, confirming the lack of contaminant cells (page 5: lines 114-116). Fourth, Interestingly, RNAseq analysis of EK cells unveils a YAP signature, according to our *in vitro* results (*HBEGF* $\log_2 = 2.52$ *CTGF* $\log_2 = 5.29$ *CYR61* $\log_2 = 7.74$ *AREG* $\log_2 = 9.64$ *TEAD4* $\log_2 = 9.49$ *YAP1* $\log_2 = 14.21$). However, we decided not to introduce these data in the text body for space limitation.

The use of EK and 266.6 cells, which are from completely different origins, as cellular models is not acceptable for screening purposes.

We understand and we are aware of the flaws brought up by this referee. For our study the use of EK was necessary, being the closest model to High grade PanIN in our hand. The impossibility of using the most appropriate control, a pure primary acinar cells from Elast-K-Ras mice, due to their prompt spontaneous transdifferentiation, is supported by literature data [I. Rooman, Y. Heremans, H. Heimberg, L. Bouwens, Modulation of rat pancreatic acinoductal transdifferentiation and expression of PDX-1 *in vitro*. *Diabetologia* **43**, 907-+ (2000). P. A. Hall, N. R. Lemoine, Rapid acinar to ductal transdifferentiation in cultured exocrine pancreas. *Journal of Pathology* **166**, 97-103 (1992)] and by shared experiences among colleagues working in this field. Moreover, extracted acinar cells usually showed low RNA quality due to their high expression of digestive enzymes, and analysis on bulk pancreas was too inaccurate, due to abundant NLGN2 expression in pancreatic insulae, nerves e vascular wall. Therefore, the use of a murine cell line instead of fresh isolated acinar cells was unavoidable. We underline that RNAseq analysis of 266-6 cell lines confirm their acinar origin showing the expression of acinar genes.

Loss of NLGN2 expression appears to be associated with EMT transition. It would be worthwhile to study the correlation between the expression of EMT markers (E-cadherin, N-cadherin, Vimentin, SNAIL, SLUG, and ZEB1) and NLGN2.

We performed Real Time PCR on mature confluent Ctrl and shNLGN2-WT-HPDE for the above-mentioned genes. Vimentin and ZEB1 are not expressed. In shNLGN2 cells, genes expressed with non-significant modulation compared to control were E-Cadherin (0,25 log₂), SNAIL (-0,35 log₂) and SLUG (0,01 log₂). The only gene significantly increased was N-Cadherin (3,01 log₂) in shNLGN2 cells. Though interesting, the increased expression of just N-Cadherin does not support the hypothesis that NLGN2 silencing induced EMT. This is also supported by the observation that in shNLGN2 cells E-cadherin was not lost even in the overgrowing structures at confluence (Fig. 4B). We think this support the idea that NLGN2 modulation happens in pre-neoplastic lesions, when cells still exhibit epithelial features. While interestingly N-Cadherin is a YAP target, further consolidating our hypothesis that NLGN2/YAP axis is relevant in low-to high grade PanINs transition, we avoided to add this data because it does not change the whole meaning of the paper.

PDAC can be classified into classical or basal-like subtypes, which perfectly explain patient survival outcomes. The association observed by the authors in the TCGA cohort may simply reflect the

loss of NLGN2 expression as the pancreatic epithelial cell undergoes dedifferentiation. Please apply the Purist classifier and examine how each subtype expresses NLGN2.

With the author's permission we used the Purist classifier and obtained the result below shown:

There was no significant difference in NLGN2 expression among classical and basal-like subtypes. Although the idea that NLGN2 might distinguish different PDAC subtypes is intriguing, it is possible that the Purist classifier approach has limitation due to considering only two PDAC subtypes. A recent finding based on single nuclei sequencing revealed a spectrum of profiles more complex than the subdivision in Classical and Basal like (Single-nucleus and spatial transcriptome profiling of pancreatic cancer identifies multicellular dynamics associated with neoadjuvant treatment” by Wang et al., Nature Genetics | VOL 54 | August 2022 | 1178–1191 | <https://doi.org/10.1038/s41588-022-01134-8>). Though it would be interesting to observe NLGN2 distribution within this sub classification, it would require an amount of work that is out form the major aim of this work.

Is there a correlation between expression of GATA6 and NLGN2?

We understand that the referee is further suggesting an association between NLGN2 expression and basal-like/classical subtype in PDAC, being GATA6 a marker for this subdivision. Having demonstrated that NLGN2 is not significantly divided among the two groups we decided to see if its expression has an effect on GATA6 expression, rather than being modulated by it. In mature confluent shNLGN2, GATA6 mRNA is slightly increased (0,45 log₂) compared to control, but this result is not significant. As mentioned above, this trend suggests a possible separation of NLGN2 subtypes, which requires further investigations according to the most updated insights into this field. Respectfully, we believe that this effort is out the main aim of the paper.

Some of the results in this study are repetitions or confirmations of previously published works, particularly those involving ColP.

We agree that the description of PALS1/PATJ complex was already reported. However, our data extend these observations highlighting the role of NLGN2 in the behavior of this complex. To our knowledge, this is the first demonstration that NLGN2 participates to PALS1/PATJ complex function. The mechanism regarding YAP activation has been validated through a series of different experiments beside ColP including phosphorylation, gene target expression, in vivo activation and rescue by YAP silencing. These data are carefully shown in Fig. 5F-J, EV5A-B, and commented in details from page 11: line 312, to page 12: line 341).

The Results section should focus solely on objective description of data. In the current version, the results are systematically introduced and discussed, making the reading lengthy and tedious. The presence of references in the Results section should be exceptional and justified.

When possible, we removed the references from the result section and moved the comments to the discussion.

To my known the aim of this study was to explain the mechanisms involved in the progression from low- to high-grade PanINs. It seems unlikely that any clues or answers to this question have been found in the end.

We agree that the most robust demonstration of our hypothesis could be obtained in a transgenic model of PDAC in which NLGN2 and YAP were differentially deleted. However, the lack of this long-time strategy does not reduce the pathogenetic value of our observation. Therefore, we respectfully argue against this referee's conclusion.

Our data are logically consequential. In vitro NLGN2 silencing induce loss of polarity and increased proliferation which are specific hallmark of high grade PanIN. YAP role is supported by the in vitro experiments showing that the phenotype induced by NLGN loss involve YAP activation, which is paralleled in vivo along with NLGN2 loss, and were rescued by silencing YAP. Mechanistically, we demonstrated that NLGN2 is necessary for the correct formation of the PATJ/PALS1 complex, a polarity regulator often altered in cancer, and NLGN2 silencing promotes the transcription of YAP target genes involved in cell survival and proliferation.

However, and in agreement with this referee's comment we added at the end of the discussion the following sentence: "a limit of this study is the lack of mouse genetics models able to demonstrate the causative connection between Nlgn2 and YAP in PanIN progression (Page 15, lines 445 - 446).

Dear Prof. Bussolino,

Thank you for the submission of your revised manuscript to our editorial offices. I have now received the reports from the two referees that I asked to re-evaluate your study, you will find below. As you will see, referee #2 now fully supports the publication of the study in EMBO reports. In contrast, referee #3 is not satisfied by the revision, but does not clearly indicate which issues remains. However, as this referee further states that s/he would nevertheless respect the decision to proceed with acceptance of the manuscript if the other referees and the editor believe that the work deserves to be published, and as referee #1 already stated in the previous round that the paper should be published, I will proceed with the manuscript.

Before formal acceptance, I have these editorial requests I ask you to address in a final revised manuscript:

- I would suggest a slightly modified title:

A Neurologin-2-YAP axis regulates progression of pancreatic intraepithelial neoplasia

- We now use CRediT to specify the contributions of each author in the journal submission system. CRediT replaces the author contribution section. Please use the free text box to provide more detailed descriptions and do NOT provide your final manuscript text file with an author contributions section. See also our guide to authors: <https://www.embopress.org/page/journal/14693178/authorguide#authorshipguidelines>

- We updated our journal's competing interests policy in January 2022 and request authors to consider both actual and perceived competing interests. Please review the policy <https://www.embopress.org/competing-interests> and update your competing interests if necessary. Please name this section 'Disclosure and Competing Interests Statement' and put it after the Acknowledgements section.

- Please reduce the number of keywords to 5 and order the manuscript sections like this, using these names:

Title page - Abstract - Keywords - Introduction - Results - Discussion - Materials and Methods - Data availability section - Acknowledgements - Disclosure and Competing Interests Statement - References - Figure legends - Tables - Expanded View Figure legends

- Please place Tables 1-3 between main and EV figure legends.

- The Data Availability section should only contain information on large datasets that have been deposited to external repositories and all access information. Please remove the statement: 'Part of the results shown in this study are based on TCGA Research Network 527 (<https://www.cancer.gov/tcga>) and on MERAV (<http://merav.wi.mit.edu>)'. Rather mention the use of TCGA and MERAV data in the relevant part(s) of the methods section.

- Per journal policy, we do not allow 'data not shown', which is stated three times in the manuscript (page 8). All data referred to in the paper should be displayed in the main or Expanded View figures, or an Appendix. Thus, please add these data (or change the text accordingly if these data are not central to the study). See: <https://www.embopress.org/page/journal/14693178/authorguide#unpublisheddata>

- Please add scale bars of similar style and thickness to all the microscopic images, using clearly visible black or white bars (depending on the background). Please place these in the lower right corner of the images themselves. Please do not write on or near the bars in the image but define the size in the respective figure legend. Presently, some of the scale bars are rather thin (or small) or hard to see against the background. Please check.

- There are magnification boxes (insets) shown in Fig. 1B. Please show in the main image where these come from and add scale bars also to the boxes.

- The bottom right image in panel EV1A seems to be a mirror image of the area marked in the box in the image above. Why is that?

- There is a magnification box (it seems) shown in Fig. EV2C (right top panel, PDAC). Please show in the main image where this comes from and add a scale bar also to the box. Finally, please describe this in the respective figure legend.

- Please make sure that the number "n" for how many independent experiments were performed, their nature (biological versus technical replicates), the bars and error bars (e.g. SEM, SD) and the test used to calculate p-values is indicated in the respective figure legends (for main, EV and Appendix figures) of the final revised manuscript. Please also check that all the p-values are explained in the legend, and that these fit to those shown in the figure. Please provide statistical testing where applicable. Please avoid the phrase 'independent experiment', but clearly state if these were biological or technical replicates. Please also indicate (e.g. with n.s.) if testing was performed, but the differences are not significant. In case n=2, please show the data as

separate datapoints without error bars and statistics. See also:

<http://www.embopress.org/page/journal/14693178/authorguide#statisticalanalysis>

If $n < 5$, please show single datapoints for diagrams. Moreover:

- Please indicate the statistical test used for data analysis in the legends of figures 1a; EV1c.
- Please note that information related to n is missing in the legends of figures 1a; EV1e.
- Please note that the error bars are not defined in the legends of figures 1a; EV1e; EV3d.
- Moreover, I would suggest to add to each legend a 'Data Information' section explaining the statistics used or providing information regarding replicates and scales. See:

- Please move all the material and methods information and the references from the Appendix to the main manuscript text file. There is no need to provide these in an Appendix. Then, please do not provide your final submission with an Appendix.

- In the reference list and in the callout, you have marked (Shaul et al, 2016) as a dataset reference. However, this seems to be a reference for a tool and not to a dataset that you have used for this study. I.e. this data citation does not refer to deposited experimental data, it seems, but refers to a journal article. Thus either turn this into a normal citation, or provide the dataset reference missing. In the Reference list, data citations must be labeled with "[DATASET]". A data reference must provide the database name, accession number/identifiers and a resolvable link to the landing page from which the data can be accessed at the end of the reference. Further instructions are available at:

- During our standard image analysis, we detected potential aberrations in the figure set, and we would like to clarify these issues. Please provide all the source data (uncropped blots) for the Western blots shown in the EV figures together with the final revised manuscript. Please upload these in one folder, but with separate files for each figure. If you make changes to the figure set, we require a further response describing what you have changed and why.

In addition, I would need from you:

Best,

Referee #2:

I support publication of this revised manuscript.

Referee #3:

I have been able to observe that the authors have attempted to integrate my comments into the new version of the manuscript. However, I have noted that, for various reasons, practically none of these comments have been fully addressed. The thesis that the authors defend is very important, and, from my point of view, it should be better justified. The working hypothesis is challenging to interpret, and there are several technical limitations that need to be considered, which has not always been the

case.

I have observed that I have been the most critical of the three reviewers. I cannot change my position as there is no new element justifying it. However, if my two colleagues and the editor of the journal believe that the work deserves to be published, I will adapt and respect that.

Dear Prof. Bussolino,

Thank you for the submission of your revised manuscript to our editorial offices. I have now received the reports from the two referees that I asked to re-evaluate your study, you will find below. As you will see, referee #2 now fully supports the publication of the study in EMBO reports. In contrast, referee #3 is not satisfied by the revision, but does not clearly indicate which issues remains. However, as this referee further states that s/he would nevertheless respect the decision to proceed with acceptance of the manuscript if the other referees and the editor believe that the work deserves to be published, and as referee #1 already stated in the previous round that the paper should be published, I will proceed with the manuscript.

Before formal acceptance, I have these editorial requests I ask you to address in a final revised manuscript:

- I would suggest a slightly modified title:

A Neuroligin-2-YAP axis regulates progression of pancreatic intraepithelial neoplasia

Done

- We now use CRediT to specify the contributions of each author in the journal submission system. CRediT replaces the author contribution section. Please use the free text box to provide more detailed descriptions and do NOT provide your final manuscript text file with an author contributions section. See also our guide to authors:

<https://www.embopress.org/page/journal/14693178/authorguide#authorshipguidelines>

Done

- We updated our journal's competing interests policy in January 2022 and request authors to consider both actual and perceived competing interests. Please review the policy <https://www.embopress.org/competing-interests> and update your competing interests if necessary. Please name this section 'Disclosure and Competing Interests Statement' and put it after the Acknowledgements section.

Done

- Please reduce the number of keywords to 5 and order the manuscript sections like this, using these names:

Title page - Abstract - Keywords - Introduction - Results - Discussion - Materials and Methods - Data availability section - Acknowledgements - Disclosure and Competing Interests Statement - References - Figure legends - Tables - Expanded View Figure legends

Done

- Please place Tables 1-3 between main and EV figure legends.

Done

- The Data Availability section should only contain information on large datasets that have been deposited to external repositories and all access information. Please remove the statement: 'Part of the results shown in this study are based on TCGA Research Network 527 (<https://www.cancer.gov/tcga>) and on MERAV (<http://merav.wi.mit.edu>)'. Rather mention the use of TCGA and MERAV data in the relevant part(s) of the methods section.

Done

- Per journal policy, we do not allow 'data not shown', which is stated three times in the manuscript (page 8). All data referred to in the paper should be displayed in the main or Expanded View figures, or an Appendix. Thus, please add these data (or change the text accordingly if these data are not central to the study). See:

<https://www.embopress.org/page/journal/14693178/authorguide#unpublisheddata>

We found only one statement in the manuscript. The others were likely removed with the first re-submission. Done.

- Please add scale bars of similar style and thickness to all the microscopic images, using clearly visible black or white bars (depending on the background). Please place these in the lower right corner of the images themselves. Please do not write on or near the bars in the image but define the size in the respective figure legend. Presently, some of the scale bars are rather thin (or small) or hard to see against the background. Please check.

Done

- There are magnification boxes (insets) shown in Fig. 1B. Please show in the main image where these come from and add scale bars also to the boxes.

Done

- The bottom right image in panel EV1A seems to be a mirror image of the area marked in the box in the image above. Why is that?

Done

- There is a magnification box (it seems) shown in Fig. EV2C (right top panel, PDAC). Please show in the main image where this comes from and add a scale bar also to the box. Finally, please describe this in the respective figure legend.

Done

- Please make sure that the number "n" for how many independent experiments were performed, their nature (biological versus technical replicates), the bars and error bars (e.g. SEM, SD) and the test used to calculate p-values is indicated in the respective figure legends (for main, EV and Appendix figures) of the final revised manuscript. Please also

check that all the p-values are explained in the legend, and that these fit to those shown in the figure. Please provide statistical testing where applicable. Please avoid the phrase 'independent experiment', but clearly state if these were biological or technical replicates. Please also indicate (e.g. with n.s.) if testing was performed, but the differences are not significant. In case n=2, please show the data as separate datapoints without error bars and statistics. See also:

<http://www.embopress.org/page/journal/14693178/authorguide#statisticalanalysis>

If n<5, please show single datapoints for diagrams. Moreover:

- Please indicate the statistical test used for data analysis in the legends of figures 1a; EV1c.

Done

- Please note that information related to n is missing in the legends of figures 1a; EV1e.

Done

- Please note that the error bars are not defined in the legends of figures 1a; EV1e; EV3d.

Done

- Moreover, I would suggest to add to each legend a 'Data Information' section explaining the statistics used or providing information regarding replicates and scales. See:

Done

- Please move all the material and methods information and the references from the Appendix to the main manuscript text file. There is no need to provide these in an Appendix. Then, please do not provide your final submission with an Appendix.

Done

- In the reference list and in the callout, you have marked (Shaul et al, 2016) as a dataset reference. However, this seems to be a reference for a tool and not to a dataset that you have used for this study. I.e. this data citation does not refer to deposited experimental data, it seems, but refers to a journal article. Thus either turn this into a normal citation, or provide the dataset reference missing. In the Reference list, data citations must be labeled with "[DATASET]". A data reference must provide the database name, accession number/identifiers and a resolvable link to the landing page from which the data can be accessed at the end of the reference. Further instructions are available

at: <http://www.embopress.org/page/journal/14693178/authorguide#referencesformat>

Done

- During our standard image analysis, we detected potential aberrations in the figure set, and we would like to clarify these issues. Please provide all the source data (uncropped blots) for the Western blots shown in the EV figures together with the final revised manuscript. Please upload these in one folder, but with separate files for each figure. If

you make changes to the figure set, we require a further response describing what you have changed and why.

We did the following changes to the figure set:

- In EV3D the “FBS” lane was shown twice, being FBS the Ctrl sample on which both Noco and HU are normalized. This was a data representation choice, driven by the sample loading order too. To avoid any misunderstanding we now show the FBS lane once. No changes were done to the quantification, which is not altered by this representation set up.
- In EV4C CACO2 panel, we originally discarded a first WB run with NLGN2 decoration due to a membrane cutting which slightly affect part of the NLGN2 band. The difference in NLGN2 expression was still appreciable though. The same samples were run on a second gel to get an “uncut” NLGN2 band, which was originally shown in EV4C. In order to match the GAPDH decoration, which was relative to the first run, we now show in EV4C the first NLGN2 decoration, and we provide the second run in the source data as a supporting replicate for NLGN2 expression.

In addition, I would need from you:

- a short, two-sentence summary of the manuscript (not more than 35 words).

NLGN2 regulates contact inhibition and epithelial polarity, which are altered in high-grade PanIN and PDAC where NLGN2 expression is lost. NLGN2 promotes PALS1/PATJ polarity complex formation which recruits YAP and reduces its function *in vitro*.

- two to four short (!) bullet points highlighting the key findings of your study (two lines each).

NLGN2 is expressed in tight junctions of normal exocrine pancreas and low-grade PanINs, and is reduced in high-grade PanINs and PDAC.

NLGN2 expression is necessary for cyst polarization and contact inhibition maintenance *in vitro*.

NLGN2 interacts with PATJ, promoting its interaction with PALS1 and YAP inactivation in confluent cells.

- a schematic summary figure as separate file that provides a sketch of the major findings (not a data image) in jpeg or tiff format (with the exact width of 550 pixels and a height of not more than 400 pixels) that can be used as a visual synopsis on our website.

I look forward to seeing the final revised version of your manuscript when it is ready.

Please let me know if you have questions regarding the revision.

Best,

Prof. Federico Bussolino
Department of Oncology; University of Turin; Candiolo, 10060, Italy; Candiolo Cancer Institute-FPO-IRCCS; Candiolo, 10060, Italy
Oncology
sp 142 Km 3,95
candiolo, Piedmont 10060
Italy

Dear Prof. Bussolino,

I am very pleased to accept your manuscript for publication in the next available issue of EMBO reports. Thank you for your contribution to our journal.

Yours sincerely,
